# Theoretically Understanding Data Reconstruction Leakage in Federated Learning

**Binghui Zhang**                                   *bzhang57@hawk.illinoistech.edu*
*Department of Computer Science*
*Illinois Institute of Technology*

**Zifan Wang**                                   *zwang248@hawk.illinoistech.edu*
*Department of Computer Science*
*Illinois Institute of Technology*

**Meng Pang**                                   *mengpang@ncu.edu.cn*
*School of Mathematics and Computer Sciences*
*Nanchang University*

**Yuan Hong**                                   *yuan.hong@uconn.edu*
*School of Computing*
*University of Connecticut*

**Binghui Wang**                                   *bwang70@illinoistech.edu*
*Department of Computer Science*
*Illinois Institute of Technology*

**Reviewed on OpenReview:** *https://openreview.net/forum?id=1UfDXeYxwk*

## Abstract

Federated learning (FL) is a collaborative learning paradigm that aims to protect data privacy. Unfortunately, recent works show FL algorithms are vulnerable to data reconstruction attacks (DRA), a serious type of privacy leakage. However, existing works lack a theoretical foundation on to what extent the devices' data can be reconstructed and the effectiveness of these attacks cannot be compared fairly due to their unstable performance. To address this deficiency, we propose a theoretical framework to understand DRAs to FL. Our framework involves bounding the data reconstruction error and an attack's error bound reflects its inherent effectiveness using Lipschitz constant. We show that a smaller Lipschitz constant indicates a stronger attacker. Under the framework, we theoretically compare the effectiveness of existing attacks (such as DLG and iDLG). We then empirically examine our results on multiple datasets, validating that the iDLG attack inherently outperforms the DLG attack.

## 1 Introduction

Federated learning (FL) (McMahan et al., 2017) has been a great potential to protect data privacy. In FL, the participating devices keep and train on their data locally, and only share the trained models, instead of the raw data, with a central server (e.g., cloud). The server updates its global model by aggregating the received device models, and broadcasts the updated global model to all participating devices such that all devices *indirectly* use all data from other devices. FL has been deployed by many companies (Google Federated Learning, 2022; Microsoft Federated Learning, 2022; IBM Federated Learning, 2022), and applied in various privacy-sensitive applications, including content suggestions for on-device keyboards (Bonawitz et al., 2019), health monitoring (Rieke et al., 2020), and medical imaging (Kaissis et al., 2020).

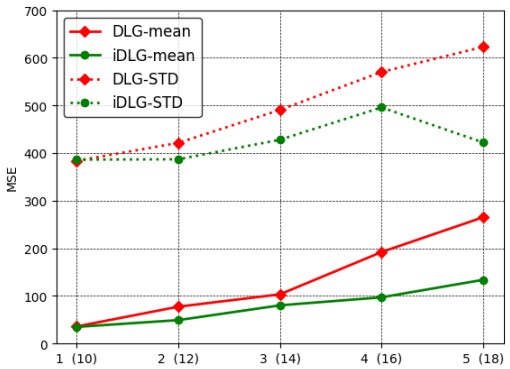

Figure 1: Impact of the initial parameters of a Gaussian distribution on the DRA performance. The x-axis marks the mean (standard deviation) of the Gaussian. The high MSE makes it difficult to conclude statistical significance in the attack outcome with empirical analysis along.

Unfortunately, recent works show that, even when device models are being shared, it is still possible for an adversary (e.g., the malicious server) to perform the severe *data reconstruction attack* (DRA) on FL (Zhu et al., 2019), where an adversary could *directly* reconstruct the device's training data via the shared device models. Later, a number of enhanced attacks (Hitaj et al., 2017; Wang et al., 2019; Zhao et al., 2020; Wei et al., 2020; Yin et al., 2021; Jeon et al., 2021; Zhu & Blaschko, 2021; Dang et al., 2021; Balunovic et al., 2022; Li et al., 2022; Fowl et al., 2022; Wen et al., 2022; Haim et al., 2022; Wu et al., 2023; Noorbakhsh et al., 2024)) are proposed by either incorporating some (known or unrealistic) prior knowledge or requiring an auxiliary dataset to simulate the training data distribution.

However, we note that existing DRA methods have several limitations: First, they are sensitive to initialization (which is also observed in (Wei et al., 2020)). For example, we show in Figure 1 we show that the attack performance of iDLG (Zhao et al., 2020) and DLG (Zhu et al., 2019) are significantly influenced by initial parameters (i.e., the mean and standard deviation) of a Gaussian distribution, from which the initial data is sampled, thus making it difficult to conclude empirically that the two attacks boast a statistically significant difference in attack outcomes. Second, existing DRAs mainly show comparison results on a FL model *at a snapshot*, which cannot reflect attacks' true effectiveness. As FL training is dynamic, an adversary can perform attacks in any stage of the training. Hence, Attack A shows better performance than Attack B at a snapshot does not imply A is truly more effective than B. Third, worse still, they lack a theoretical understanding on to what extent the training data can be reconstructed. These limitations make existing DRAs not be easily and fairly compared and hence it is unknown which attacks are inherently more effective.

In this paper, we aim to bridge the gap and propose a theoretical framework to understand DRAs to FL. Specifically, our framework bounds the error between the private data and the reconstructed counterpart in the whole FL training, where an attack's (smaller) error bound reflects its inherent (better) attack effectiveness. Our theoretical results show that when an attacker's reconstruction function has a smaller Lipschitz constant, this attack intrinsically performs better[1]. Under the framework, we can theoretically compare the existing DRAs by directly comparing their bounded errors. We test our framework on multiple attacks and benchmark datasets. E.g., our results show InvGrad (Geiping et al., 2020) performs better than DLG (Zhu et al., 2019) and iDLG (Zhao et al., 2020) on complex datasets, while iDLG is comparable or slightly better than DLG.

## 2 Preliminaries and Problem Setup

### 2.1 Federated Learning (FL)

The FL paradigm enables a server to coordinate the training of multiple local devices through multiple rounds of global communications, without sharing the local data. Suppose there are $N$ devices and a central server participating in FL. Each $k$-th device owns a training dataset $D^k = \{(\mathbf{x}_j^k, y_j^k)\}_{j=1}^{n_k}$ with $n_k$ samples, and each sample $\mathbf{x}_j^k$ has a label $y_j^k$. FL considers the following distributed optimization problem:

---

[1]Under realistic attack scenarios where an ideal model exists. For instance, a constant function has $L_{\mathcal{R}} = 0$, but it will have high first term in 3, as the ideal reconstruction function will have extremely high MSE.

$$\min_{\mathbf{w}} \mathcal{L}(\mathbf{w}) = \sum_{k=1}^{N} p_k \mathcal{L}_k(\mathbf{w}), \tag{1}$$

where $p_k \geq 0$ is the weight of the $k$-th device and $\sum_{k=1}^{N} p_k = 1$; and the $k$-th device's local objective is:

$$\mathcal{L}_k(\mathbf{w}) = \frac{1}{n_k} \sum_{j=1}^{n_k} \ell(\mathbf{w}; (\mathbf{x}_j^k, y_j^k))$$

with $\ell(\cdot; \cdot)$ an algorithm-dependent loss function.

**FedSGD** is the *de facto* FL algorithm to solve Equation (1) in an iterative way. In each communication round, each $k$-th device only shares the gradients $\nabla_{\mathbf{w}} \mathcal{L}_k(\mathbf{w})$ instead of the raw data $D^k$ to the server McMahan et al. (2017). Specifically, in the current round $t$, each $k$-th device first downloads the latest global model (denoted as $\mathbf{w}_{t-1}$) from the server and initializes its local model as $\mathbf{w}_t^k = \mathbf{w}_{t-1}$; then it performs (e.g., $E$) local SGD updates as below:

$$\mathbf{w}_{t+j}^k \leftarrow \mathbf{w}_{t+j-1}^k - \eta_{t+j} \nabla \mathcal{L}_i(\mathbf{w}_{t+j}^k; \xi_{t+j}^k), j = 1, \cdots, E$$

where $\eta_{t+j}$ is the learning rate and $\xi_{t+j}^k$ is sampled from the local data $D^k$ uniformly at random.

Next, the server updates the global model $\mathbf{w}_t$ by aggregating full or partial device models. The final global model is downloaded by all devices for their learning tasks.

- *Full device participation.* It requires all device models for aggregation, and the server performs $\mathbf{w}_t \leftarrow \sum_{k=1}^{N} p_k \mathbf{w}_t^k$ with $p_k = \frac{n_k}{\sum_{i=1}^{N} n_i}$ and $\mathbf{w}_t^k = \mathbf{w}_{t+E}^k$. Note that full device participation means the server must wait for the slowest devices, which is often unrealistic in practice.

- *Partial device participation.* This is a more realistic setting as it does not require the server to know all device models. Suppose the server only needs $K$ ($< N$) device models for aggregation and discards the remaining ones. Let $\mathcal{S}_t$ be the set of $K$ chosen devices in the $t$-th iteration. Then, the server's aggregation step performs $\mathbf{w}_t \leftarrow \frac{N}{K} \sum_{k \in \mathcal{S}_t} p_k \mathbf{w}_t^k$ with $\mathbf{w}_t^k = \mathbf{w}_{t+E}^k$.

**Quantifying the degree of non-IID (heterogeneity):** Real-world FL applications often do not satisfy the IID assumption for data among local devices. Li et al. (2020) proposed a way to quantify the degree of non-IID. Specifically, let $\mathcal{L}^*$ and $\mathcal{L}_k^*$ be the minimum values of $\mathcal{L}$ and $\mathcal{L}_k$, respectively. Then, the term $\Gamma = \mathcal{L}^* - \sum_{k=1}^{N} p_k \mathcal{L}_k^*$ is used for quantifying the degree of non-IID. If the data are IID, then $\Gamma$ obviously goes to zero as the number of samples grows. If the data are non-IID, then $\Gamma$ is nonzero, and its magnitude reflects the heterogeneity of the data distribution.

## 2.2 Optimization-based DRAs to FL

Existing DRAs assume an honest-but-curious server, i.e., the server has access to all device models in all communication rounds, follows the FL protocol, and aims to infer devices' private data. Given the private data $\mathbf{x} \in [0, 1]^d$ with private label $y^2$, we denote the reconstructed data by a malicious server as $(\hat{\mathbf{x}}, \hat{y}) = \mathcal{R}(\mathbf{w}_t)$, where $\mathcal{R}(\cdot)$ indicates a *data reconstruction function*, and $\mathbf{w}_t$ can be any intermediate global model. Modern optimization-based DRAs use different $\mathcal{R}(\cdot)$, but mainly based on *gradient matching*. Specifically, they solve the below optimization problem:

$$\mathcal{R}(\mathbf{w}_t) = \arg \min_{\mathbf{x}' \in [0,1]^d, y'} \text{GML}(g_{\mathbf{w}_t}(\mathbf{x}, y), g_{\mathbf{w}_t}(\mathbf{x}', y')) + \lambda \text{Reg}(\mathbf{x}') \tag{2}$$

where we denote the gradient of loss w.r.t. $(\mathbf{x}, y)$ be $g_{\mathbf{w}_t}(\mathbf{x}, y) = \nabla_{\mathbf{w}} \mathcal{L}(\mathbf{w}_t; (\mathbf{x}, y))$ for simplicity of notation. $\text{GML}(\cdot, \cdot)$ means the *gradient matching loss* (i.e., the distance between the real gradients and estimated gradients) and $\text{Reg}(\cdot)$ is an auxiliary *regularizer* for the reconstruction. Here, we list $\text{GML}(\cdot, \cdot)$ and $\text{Reg}(\cdot)$ for three representative DRAs, and more attacks are shown in Appendix D.1.2.

[2]This can be a single data sample or a batch of data samples.

---

**Algorithm 1** Iterative solvers for optimization-based DRAs

---

**Input:** Model parameters $\mathbf{w}_t$; true gradient $g(\mathbf{x}, y)$; $\eta, \lambda$.
**Output:** Reconstructed data $\hat{\mathbf{x}}$.

---

1: Initialize dummy input(s) $\mathbf{x}'_0$ from a Gaussian distribution, and dummy label(s) $y'_0$
2: **for** $i = 0; i < I; i + + $ **do**
3:      $\mathrm{L}(\mathbf{x}'_i) = \mathrm{GML}(g_{\mathbf{w}_t}(\mathbf{x}, y), g_{\mathbf{w}_t}(\mathbf{x}'_i, y'_i)) + \lambda \mathrm{Reg}(\mathbf{x}'_i)$
4:      $\mathbf{x}'_{i+1} \leftarrow \mathrm{SGD}(\mathbf{x}'_i; \theta^i) = \mathbf{x}'_i - \eta \nabla_{\mathbf{x}'_i} \mathrm{L}(\mathbf{x}'_i)$
5:      $\mathbf{x}'_{i+1} = \mathrm{Clip}(\mathbf{x}'_{i+1}, 0, 1)$
6: **end for**
7: **return** $\mathbf{x}'_I$

---

- **DLG** (Zhu et al., 2019) uses the mean squared error as the gradient matching loss, i.e., $\mathrm{GML}(g_{\mathbf{w}_t}(\mathbf{x}, y), g_{\mathbf{w}_t}(\mathbf{x}', y')) = \|g_{\mathbf{w}_t}(\mathbf{x}, y) - g_{\mathbf{w}_t}(\mathbf{x}', y')\|_2^2$ and does not use a regularizer.

- **iDLG** (Zhao et al., 2020) estimates the label $y$ before solving Equation (2). iDLG solves $\hat{\mathbf{x}} = \arg\min_{\mathbf{x}'} \mathbb{E}_{\mathbf{x}}[\mathrm{GML}(g_{\mathbf{w}_t}(\mathbf{x}, y), g_{\mathbf{w}_t}(\mathbf{x}', \hat{y})) + \lambda \mathrm{Reg}(\mathbf{x}')]$, assuming the estimated label is $\hat{y}$, where it uses the same $\mathrm{GML}(\cdot)$ as DLG and also does not have a regularizer.

- **InvGrad** (Geiping et al., 2020) improves upon DLG and iDLG. It first estimates the private label as $\hat{y}$ in advance. Then it uses a negative cosine similarity as $\mathrm{GML}(\cdot)$ and a total variation regularizer $\mathrm{Reg}_{\mathrm{TV}}(\cdot)$ as an image prior. Specifically, InvGrad solves for $\hat{\mathbf{x}} = \arg\min_{\mathbf{x}'} \mathbb{E}_{\mathbf{x}}[1 - \frac{\langle g_{\mathbf{w}_t}(\mathbf{x}, y), g_{\mathbf{w}_t}(\mathbf{x}', \hat{y})\rangle}{\|g_{\mathbf{w}_t}(\mathbf{x}, y)\|_2 \cdot \|g_{\mathbf{w}_t}(\mathbf{x}', \hat{y})\|_2} + \lambda \mathrm{Reg}_{\mathrm{TV}}(\mathbf{x}')]$.

Algorithm 1 shows the pseudo-code of iterative solvers for DRAs and Algorithm 4 in Appendix shows more details for each attack. As the label $y$ can be often accurately inferred, we now only consider reconstructing the data $\mathbf{x}$ for notation simplicity. Then, the attack performance is measured by the similarity $\mathrm{sim}(\hat{\mathbf{x}}, \mathbf{x})$ between $\hat{\mathbf{x}}$ and $\mathbf{x}$. The larger similarity, the better attack performance. In the paper, we use the common similarity metric, i.e., the negative mean-square-error $\mathrm{sim}(\hat{\mathbf{x}}, \mathbf{x}) = -\mathbb{E}\|\hat{\mathbf{x}} - \mathbf{x}\|^2$, where the expectation $\mathbb{E}$ considers the randomness during reconstruction.

## 3 A Theoretical Framework to Understand DRAs to Federated Learning

Though many DRAs to FL have been proposed, it is still unknown how to *theoretically* compare the effectiveness of existing attacks, as stated in Introduction. In this section, we understand DRAs to FL from a theoretical perspective. We first derive a reconstruction error bound for convex objective losses. The error bound involves knowing the Lipschitz constant of the data reconstruction function. Directly calculating the exact Lipschitz constant is computationally challenging. We then adapt existing methods to calculate its upper bound. We argue that an attack with a smaller upper bound is intrinsically more effective. We also emphasize our theoretical framework is applicable to any adversary who knows the global model during FL training (see below Theorem 3.2 and Theorem 3.3).

### 3.1 Bounding the Data Reconstruction Error

Given random data $\mathbf{x}$ from a device, our goal is to bound the common norm-based reconstruction error[3], i.e., $\mathbb{E}\|\mathbf{x} - \mathcal{R}(\mathbf{w}_t)\|^2$ at any round $t$, where $\mathcal{R}(\cdot)$ can be any DRA and the expectation considers the randomness in $\mathcal{R}(\cdot)$, e.g., due to different initializations. Directly bounding this error is challenging because the global model dynamically aggregates local device models, which are trained by a (stochastic) learning algorithm and whose learning procedure is hard to characterize during training. To alleviate this issue, we introduce the optimal global model $\mathbf{w}^\star$ that can be learnt by the FL algorithm. Then, we can bound the error as follows:

$$\mathbb{E}\|\mathbf{x} - \mathcal{R}(\mathbf{w}_t)\|^2 = \mathbb{E}\|\mathbf{x} - \mathcal{R}(\mathbf{w}^\star) + \mathcal{R}(\mathbf{w}^\star) - \mathcal{R}(\mathbf{w}_t)\|^2$$
$$\leq 2(\mathbb{E}\|\mathbf{x} - \mathcal{R}(\mathbf{w}^\star)\|^2 + \mathbb{E}\|\mathcal{R}(\mathbf{w}^\star) - \mathcal{R}(\mathbf{w}_t)\|^2). \tag{3}$$

---

[3]The norm-based mean-square-error (MSE) bound can be easily generalized to the respective PSNR bound. This is because PSNR = -10 log (MSE). However, it is unable to generalize to SSIM or LPIPS since these metrics do not have an analytic form.

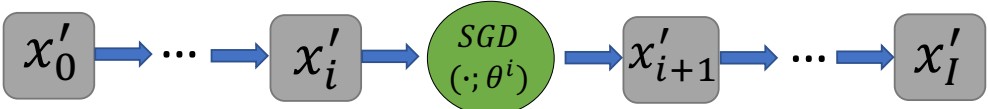

Figure 2: Iterative solvers for DRAs as unrolled deep feed-forward networks. We map the $i$-th SGD iteration in DRAs into a single network layer, and stack $I$ layers to form an $I$-layer deep network. Feeding forward data through the $I$-layer network is equivalent to executing $I$ SGD updates. The trainable parameters $\{\theta^i\}$ are colored in blue in Algorithm 1, and $\theta^i$ connects the $i$-th layer and $i+1$-th layer. These parameters can be learned from intermediate reconstructed data $\mathbf{x}'_i$ by training the deep feed-forward network.

Note that the first term in Equation (3) is a constant and can be directly computed under a given reconstruction function $\mathcal{R}(\cdot)$ and a convex loss used in FL. Specifically, if the loss in each device is convex, the global model can converge to the *optimal* $\mathbf{w}^*$ based on theoretical results in (Li et al., 2020). Then we can obtain $\mathcal{R}(\mathbf{w}^*)$ per attack and compute the first term. In our experiments, we run the FL algorithm until the loss difference between two consecutive iterations is less than $1e-5$, and treat the final global model as $\mathbf{w}^*$.

Now our goal reduces to bounding the second term. However, it is still challenging without knowing any property of the reconstruction function $\mathcal{R}(\cdot)$. In practice, we note $\mathcal{R}(\cdot)$ is often Lipschitz continuous, which can be verified later.

**Proposition 3.1.** $\mathcal{R}(\cdot)$ *is* $L_{\mathcal{R}}$*-Lipschitz continuous: there exists a constant* $L_{\mathcal{R}}$ *such that* $\|\mathcal{R}(\mathbf{v}) - \mathcal{R}(\mathbf{w})\| \le L_{\mathcal{R}}\|\mathbf{v} - \mathbf{w}\|, \forall \mathbf{v}, \mathbf{w}$. *The smallest* $L_{\mathcal{R}}$ *is called the* Lipschitz constant.

Next, we present our theoretical results and their proofs are seen in Appendix B and Appendix C.

**Theoretical results with full device participation:** We first analyze the case where all devices participate in the aggregation on the server in each communication round. Assume the **FedAvg** algorithm stops after $T$ iterations and returns $\mathbf{w}_T$ as the solution. Let $L, \mu, \sigma_k, G$ be defined in Assumptions A.1-A.4 (more details in Appendix A) and $L_{\mathcal{R}}$ be defined in Proposition 3.1.

**Theorem 3.2.** *Let Assumptions A.1-4 hold. Choose* $\gamma = \max\{8L/\mu, E\}$ *and learning rate* $\eta_t = \frac{2}{\mu(\gamma+t)}$. *Let* $B = \sum_{k=1}^{N} p_k^2 \sigma_k^2 + 6L\Gamma + 8(E-1)^2 G^2$. *Then, for any round* $t$, *FedAvg with full device participation satisfies*

$$\mathbb{E}\|\mathbf{x} - \mathcal{R}(\mathbf{w}_t)\|^2 \le 2\mathbb{E}\|\mathbf{x} - \mathcal{R}(\mathbf{w}^\star)\|^2 + \frac{2L_{\mathcal{R}}^2}{\gamma+t}\Big(\frac{4B}{\mu^2} + (\gamma+1)\|\boldsymbol{w}_1 - \mathbf{w}^*\|^2\Big). \tag{4}$$

**Theoretical results with partial device participation:** As discussed in Section 2, partial device participation is more practical. Recall that $\mathcal{S}_t$ contains the $K$ active devices in the $t$-th iteration. To show our theoretical results, we require the $K$ devices in $\mathcal{S}_t$ are selected from a distribution (e.g., $p_1, p_2, \cdots, p_N$) independently and with replacement, following (Sahu et al., 2018; Li et al., 2020). Then FedAvg performs aggregation as $\mathbf{w}_t \leftarrow \frac{1}{K}\sum_{k \in \mathcal{S}_t} \mathbf{w}_t^k$.

**Theorem 3.3.** *Let Assumptions A.1-A.4 hold. Let* $\gamma, \eta_t$, *and* $B$ *be defined in Theorem 3.2, and define* $C = \frac{4}{K}E^2 G^2$. *For any round* $t$, *FedAvg with* $\mathcal{S}_t$ *device participation satisfies*

$$\mathbb{E}\|\mathbf{x} - \mathcal{R}(\mathbf{w}_t)\|^2 \le 2\mathbb{E}\|\mathbf{x} - \mathcal{R}(\mathbf{w}^\star)\|^2 + \frac{2L_{\mathcal{R}}^2}{\gamma+t}\Big(\frac{4(B+C)}{\mu^2} + (\gamma+1)\|\boldsymbol{w}_1 - \mathbf{w}^*\|^2\Big). \tag{5}$$

### 3.2 Computing the Lipschitz Constant for Data Reconstruction Functions

In this part, we show how to calculate the Lipschitz constant for data reconstruction function. Our idea is to first use the strong connection between optimizing DRAs and the corresponding unrolled deep neural networks; and then adapt existing methods to approximate the Lipschitz upper bound.

**Iterative solvers for optimization-based DRAs as unrolled deep feed-forward networks:** Recent works Chen et al. (2018); Li et al. (2019); Monga et al. (2021) show a strong connection between iterative algorithms and deep network architectures. The general idea is: starting with an abstract iterative algorithm,

---

**Algorithm 2** AutoLip

---

**Require:** function $f$ and its computation graph $(g_1, ..., g_H)$
**Ensure:** Lipschitz upper bound $L_{AutoLip} \geq L_f$
1: $\phi_0(\mathbf{x}) = \mathbf{x}$; $\phi_h(\mathbf{x}) = f(\mathbf{x})$
2: $\phi_h(\mathbf{x}) = g_h(\mathbf{x}, \phi_1(\mathbf{x}), \cdots, \phi_{h-1}(\mathbf{x})), \forall h \in [1, H]$
3: $\mathcal{Z} = \{(z_0, ..., z_H) : \forall h \in [0, H], \phi_h \text{ is constant} \Rightarrow z_h = \phi_h(0)\}$
4: $L_0 \leftarrow 1$
5: **for** $h = 1$ to $H$ **do**
6: $\quad L_h \leftarrow \sum_{i=1}^{h-1} \max_{z \in \mathcal{Z}} \|\partial_i g_h(z)\|_2 L_i$
7: **end for**
8: **return** $L_{AutoLip} = L_H$

---

**Algorithm 3** Power method to calculate the matrix $\ell_2$-norm

---

**Require:** affine function $f : \mathbb{R}^n \to \mathbb{R}^m$, #iterations $Iter$
**Ensure:** Upper bound of the Lipschitz constant $L_f$
1: **for** $j = 1$ to $Iter$ **do**
2: $\quad v \leftarrow \nabla g(v)$ where $g(x) = \frac{1}{2}\|f(x) - f(0)\|_2^2$
3: $\quad \lambda \leftarrow \|v\|_2$
4: $\quad v \leftarrow v/\lambda$
5: **end for**
6: **return** $L_f = \|f(v) - f(0)\|_2$

---

we map one iteration into a single network layer, and stack a finite number of (e.g., $H$) layers to form a deep network, which is also called *unrolled deep network*. Feeding the data through an $H$-layer network is hence equivalent to executing the iterative algorithm $H$ iterations. The parameters of the unrolled networks are learnt from data by training the network in an end-to-end fashion. From Algorithm 1, we can see the trajectory of an iterative solver for an optimization-based DRA corresponds to a customized unrolled deep feed-forward network. Specifically, we treat the initial $\mathbf{x}'_0$ (and $\mathbf{w}_t$) as the input, the intermediate reconstructed $\mathbf{x}'_i$ as the $i$-th hidden layer, followed by a nonlinear clip function, and the final reconstructed data $\hat{\mathbf{x}} = \mathbf{x}'_I$ as the output of the network. Given intermediate $\{\mathbf{x}'_i\}$ with a set of data samples, we can train parameterized deep feed-forward networks (universal approximation) to fit them, e.g., via the greedy layer-wise training strategy Bengio et al. (2006), where each $\theta^i$ means the model parameter connecting the $i$-th layer and $i+1$-th layer. Figure 2 visualizes the unrolled deep feed-forward network for the optimization-based DRA.

**Definition 3.4** (Deep Feed-Forward Network)**.** An $H$-layer feed-forward network is an function $T_{MLP}(\mathbf{x}) = f_H \circ \rho_{H-1} \circ \cdots \circ \rho_1 \circ f_1(\mathbf{x})$, where $\forall h$, the $h$-th hidden layer $f_h : \mathbf{x} \mapsto \mathbf{M}_h\mathbf{x} + b_h$ is an affine function and $\rho_h$ is a non-linear activation function.

**Upper bound Lipschitz computation:** Virmaux & Scaman (2018) showed computing the exact Lipschitz constant for deep (even 2-layer) feed-forward networks is NP-hard. Hence, they resort to an approximate computation and propose a method called **AutoLip** to obtain an upper bound of the Lipschitz constant. AutoLip relies on the concept of *automatic differentiation* Griewank & Walther (2008), a principled approach that computes differential operators of functions from consecutive operations through a computation graph. When the operations are all locally Lipschitz-continuous and their partial derivatives can be computed and maximized, AutoLip can compute the Lipschitz upper bound efficiently. Algorithm 2 shows the details.

With Autolip, Virmaux & Scaman (2018) showed that a feed-forward network with 1-Lipschitz activation functions has an upper bounded Lipschitz constant below.

**Lemma 3.5.** *For any $H$-layer feed-forward network with $1$-Lipschitz activation functions, the AutoLip upper bound becomes $L_{AutoLip} = \prod_{h=1}^{H} \|\mathbf{M}_h\|_2$, where $\mathbf{M}_h$ is defined in Definition 3.4.*

Note that a matrix $\ell_2$-norm equals to its largest singular value, which could be computed efficiently via the *power method* (Mises & Pollaczek-Geiringer, 1929). More details shown in Algorithm 3 (A better estimation algorithm leads to a tighter Lipschitz bound). The used Clip activation function is 1-Lipschitz. Hence, by applying Lemma 3.5 to the optimization-based DRAs, we can derive an upper bounded Lipschitz $L_{\mathcal{R}}$.

## 4 Evaluation

### 4.1 Experimental Setup

**Datasets and models:** We conduct experiments on three benchmark image datasets, i.e., MNIST (LeCun, 1998), Fashion-MNIST(FMNIST) (Xiao et al., 2017), and CIFAR10 (Krizhevsky et al., 2009). We examine our theoretical results on the FL algorithm that uses the $\ell_2$-regularized logistic regression ($\ell_2$-LogReg) and

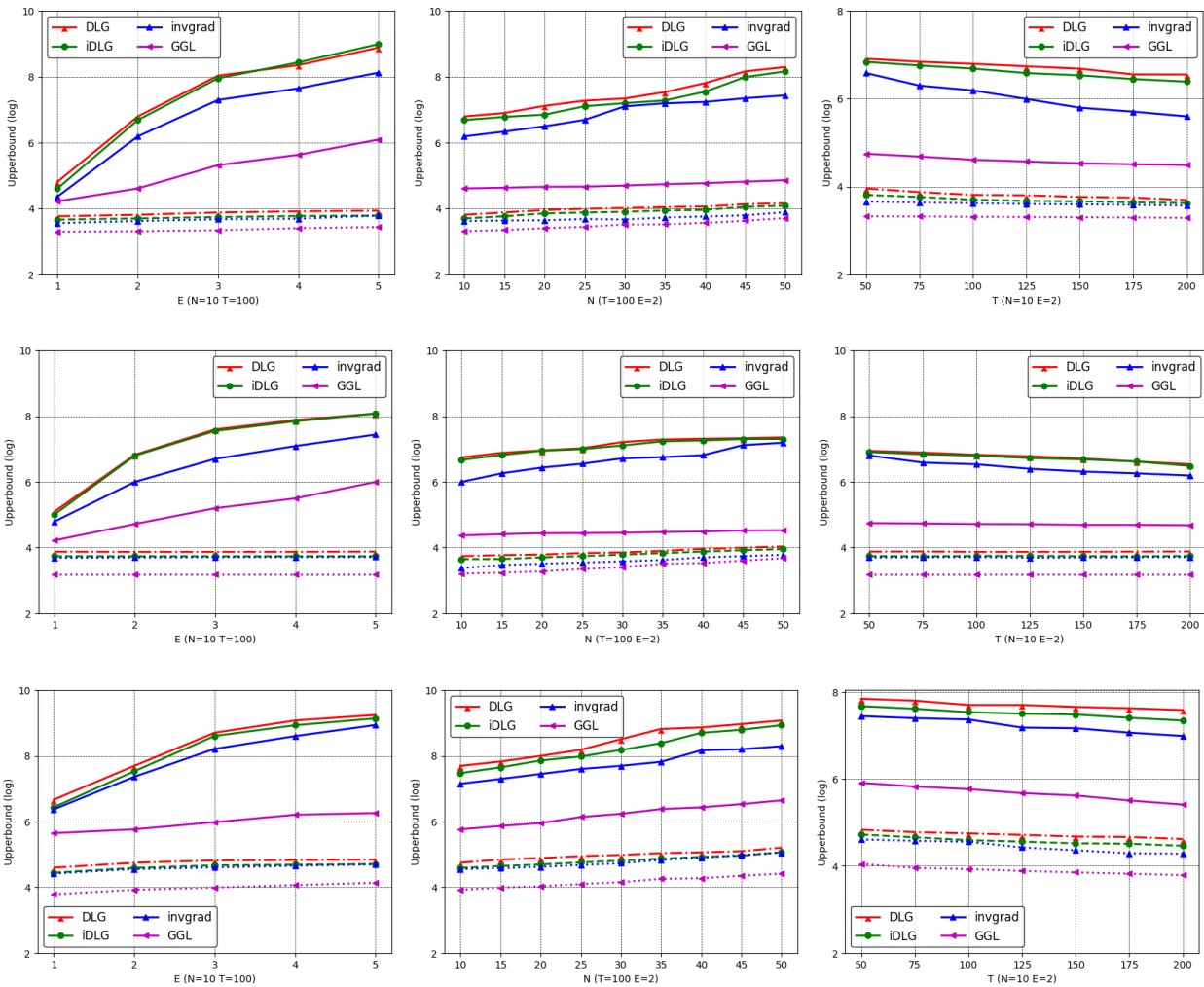

Figure 3: Results of federated $\ell_2$-LogReg on MNIST (Row 1) FMNIST (Row 2) CIFAR10 (Row 3), impact of E, N, T, left to right —single image recovery. **Dashed lines** are *average* empirical reconstruction errors obtained by existing DRAs, while **solid lines** are *upper bound* errors obtained by our theoretical results. Y-axis is in a log scale. We observe that iDLG slightly outperforms DLG both empirically and theoretically; a larger $E$ and $N$ incur larger upper bound error, while a larger $T$ generates a smaller upper bound error. Empirical attacks are insensitive to these parameters.

convex 2-layer linear convolutional network (2-LinConvNet) (Pilanci & Ergen, 2020), since their loss functions satisfy Assumptions A.1-A.4. In the experiments, we evenly distribute the training data among the $N$ devices. Based on this setting, we can calculate $L, \mu, \sigma_k$, and $G$ used in our theorems, respectively. In addition, we can compute the Lipschitz constant $L_\mathcal{R}$ via the unrolled feed-forward network. These values together are used to compute the upper bound of our Theorems 3.2 and 3.3. *More details about the two algorithms, unrolled feed-forward network, and calculation of parameter values shown in Appendix D.1.*

**Attack baselines:** We test our theoretical results on four optimization-based data reconstruction attacks, i.e., DLG (Zhu et al., 2019), iDLG (Zhao et al., 2020), InvGrad (Geiping et al., 2020), and the GGL (Li et al., 2022). The algorithms and descriptions of these attacks are deferred to Appendix D.1. We test these attacks on recovering both the single image and a batch of images in each device.

**Parameter setting:** Several important hyperparameters in the FL training would affect our theoretical results: the total number of devices $N$, the total number of global rounds $T$, and the number of local SGD

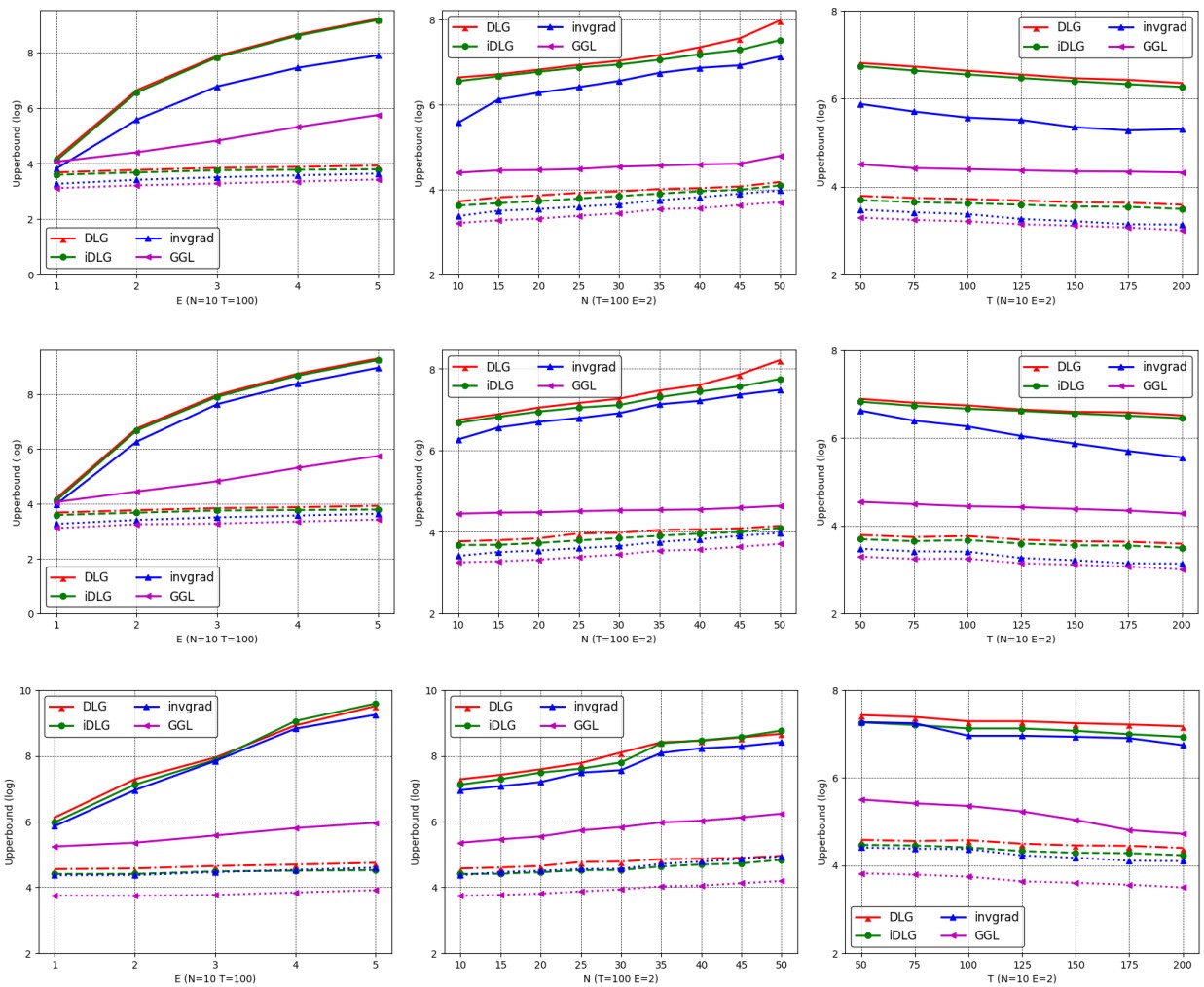

Figure 4: Results of federated 2-LinConvNet on MNIST (Row 1), FMNIST (Row 2), CIFAR10 (Row 3), impact of E, N, T from left to right—single image recovery.

updates $E$. By default, we set $T = 100$ and $E = 2$. We set $N = 10$ on the three datasets for the single image recovery, while set $N = 15, 10, 5$ on the three datasets for the batch images recovery, considering their different difficulty levels. we consider full device participation. When studying the impact of these hyperparameters, we fix others as the default value.

## 4.2 Experimental Results

In this section, we test the upper bound reconstruction error by our theoretical results for single image and batch images recovery. We also show the *average* (across 10 iterations) reconstruction errors that are empirically obtained by the baseline DRAs with different initializations. The *best* possible (one-snapshot) empirical results of the baseline attacks are also reported in Table 1 in Appendix D.2.

### 4.2.1 Results on single image recovery

Figure 3-Figure 5 show the single image recovery results on the three datasets and two FL algorithms, respectively. We have several observations. *First*, iDLG has smaller upper bound errors than DLG, indicating iDLG outperforms DLG intrinsically. One possible reason is that iDLG can accurately estimate the labels,

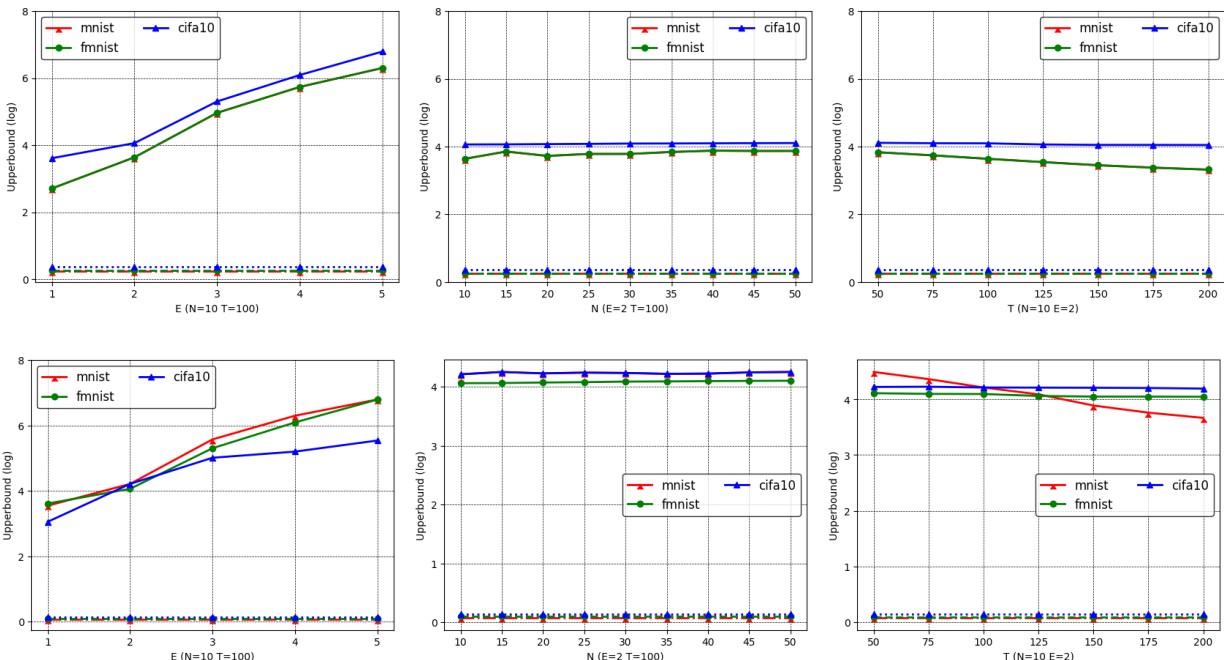

Figure 5: Results of federated $\ell_2$-LogReg (Row 1) and 2-LinConvNet (Row 2) on Robbing, impact of E, N, T left to right—single image recovery. We observe that Robbing has much smaller bounded errors and is even smaller than GGL (See Figure 3). This is because the equation solving used by Robbing is accurate on the simple federated $\ell_2$-LogReg model that uses a linear layer.

which ensures data reconstruction to be more stable. Such a stable reconstruction yields a smaller Lipschitz $L_{\mathcal{R}}$, and thus a smaller upper bound in Theorem 3.2. InvGrad is on top of iDLG and adds a TV regularizer, and obtains smaller error bounds than iDLG. This implies the TV prior can help stabilize the data reconstruction and hence is beneficial for reducing the error bounds. Note that the error bounds of these three attacks are (much) larger than the average empirical errors, indicating there is still a gap between empirical results and theoretical results. Further, GGL has (much) smaller bounded errors than DLG, iDLG, and InvGrad. This is because GGL trains an encoder on the *whole* dataset to learn the image manifold, and then uses the encoder for data reconstruction, hence producing smaller $L_{\mathcal{R}}$. In Figure 3(b), we observe its bounded error is close to the empirical error. For instance, we calculate that the estimated $L_{\mathcal{R}}$ for the DLG, iDLG, InvGrad, and GGL attacks in the default setting on MNIST are 22.17, 20.38, 18.43, and 13.36, respectively.

*Second*, the error bounds are consistent with the *average* empirical errors, validating they have a *strong* correlation. Particularly, we calculate that the Pearson correlation coefficients between the error bound and the averaged empirical error on these attacks in *all* settings are larger than 0.9.

*Third*, the error bounds do not show consistent correlations with the *best* empirical errors. For instance, we can see GGL has lower error bounds than InvGrad on $\ell_2$-LogReg, but its *best* empirical error is larger than InvGrad on MNIST (3.14 vs 3.09). Similar observations on CIFAR10 on federated 2-LinConvNet, where InvGrad's error bound is smaller than iDLG's, but its best empirical error is larger (4.33 vs 3.37). More details see Table 1 in Appendix D.2. The reason is that the reported empirical errors are the best possible *one-snapshot* results with a certain initialization, which do not reflect the attacks' inherent effectiveness. Recall in Figure 1 that empirical errors obtained by these attacks could be sensitive to different initializations. In practice, the attacker may need to try many initializations (which could be time-consuming) to obtain the best empirical error.

**Impact of $E$, $N$, and $T$**: When the local SGD updates $E$ and the number of total clients $N$ increase, the upper bound error also increases. This is because a large $E$ and $N$ will make FL training unstable and hard

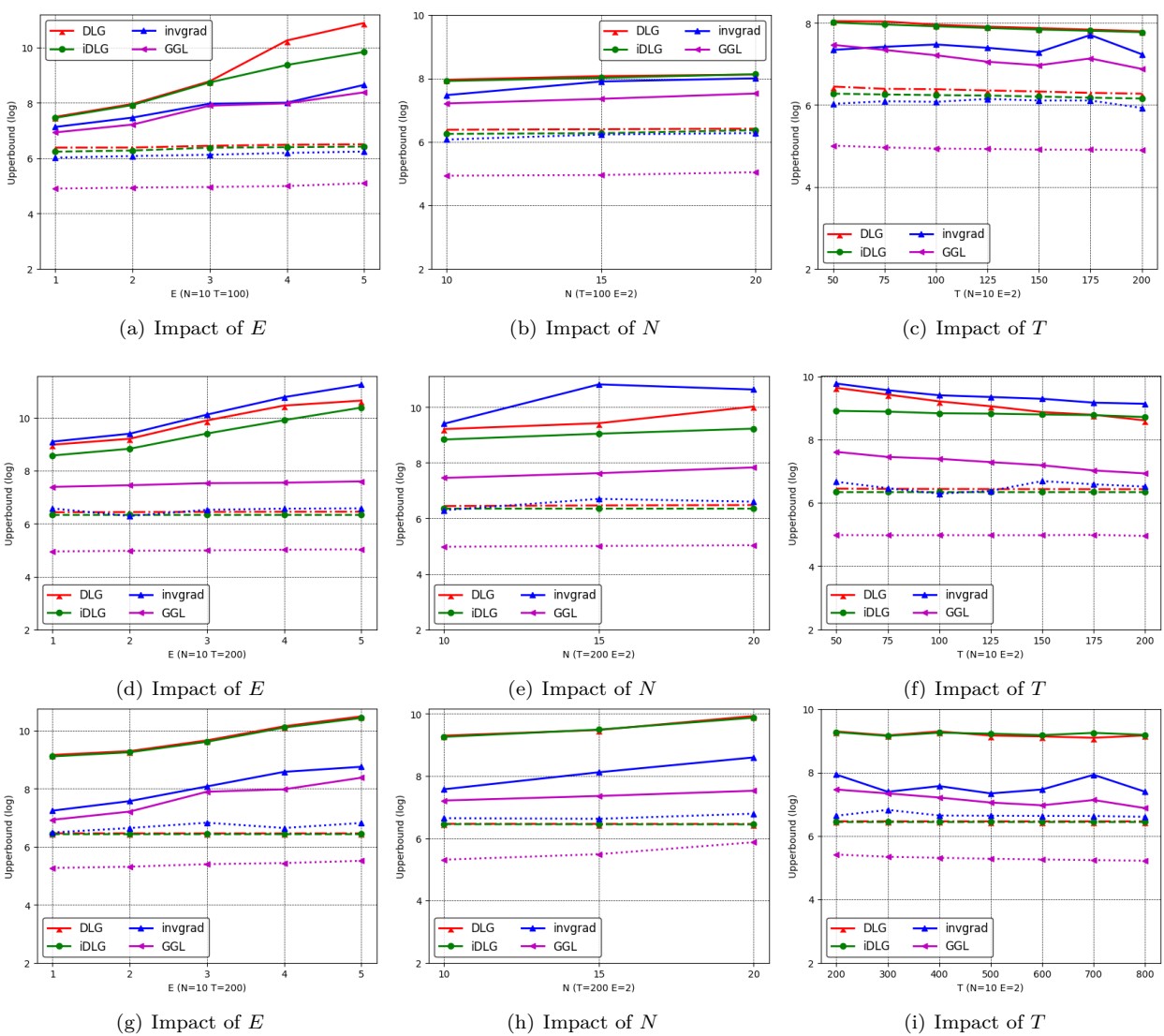

Figure 6: Results of federated $\ell_2$-LogReg on MNIST(a-c), FMNIST(d-f) CIFAR10(g-i)—batch images recovery.

to converge, as verified in (Li et al., 2020). On the other hand, a larger total number of global rounds $T$ tends to produce a smaller upper bounded error. This is because a larger $T$ stably makes FL training closer to the global optimal under convex loss.

### 4.2.2 Results on batch images recovery

Figures 8-10 in Appendix D.2 show the results of batch images recovery on the three image datasets. As federated 2-LinConvNet has similar trends, we only show federated $\ell_2$-LogReg results for simplicity. Our key observations are: First, similar to results on single image recovery, GGL performs the best; InvGrad outperforms iDLG, which outperforms DLG both empirically and theoretically. Moreover, a larger $E$ and $N$ incur larger upper bound error, while a larger $T$ generates smaller upper bound error. Second, both empirical errors and upper bound errors for batch images recovery are much larger than those for single image recovery. This indicates that batch images recovery are more difficult than single image recovery, as validated in many existing works such as (Geiping et al., 2020; Yin et al., 2021).

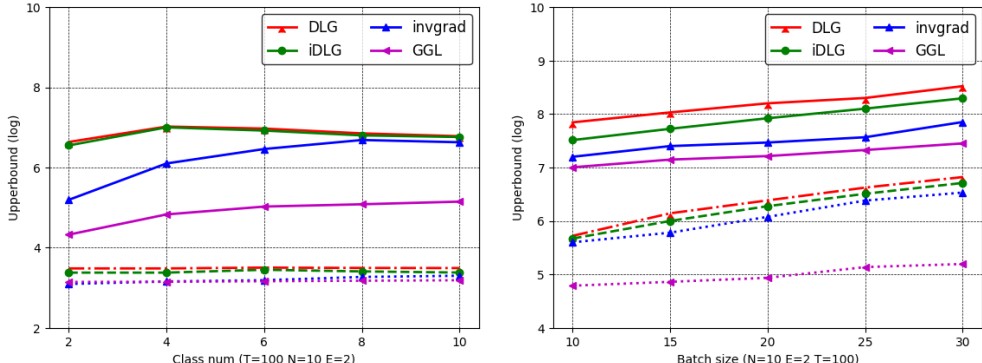

Figure 7: Impact of #classes and batch size on MNIST.

## 5 Discussion

**First term vs second term in the error bound 3:** We calculate the first term and second term of the error bound in our theorem in the default setting. The two terms in DLG, iDLG, InvGrad, and GGL on MNIST are (30.48, 396.72), (25.25, 341.10), (22.06, 218.28), and (20.21, 29.13) respectively. The second term dominates the error bound with significantly higher value.

**Error bounds vs. degree of non-IID:** The non-IID of data across clients can be controlled by the number of classes per client—small number indicates a larger degree of non-IID. Here, we tested #classes=2, 4, 6, 8 on MNIST and the results are shown in Figure 7(a). We can see the bounded errors are relatively stable vs. #classes on DLG, iDLG, and GGL, while InvGrad has a larger error as the #classes increases. The possible reason is that DLG and iDLG are more stable than InvGrad, which involves a more complex optimization.

**Error bounds vs. batch size:** Our batch results use a batch size 20. Here, we also test batch size=10, 15, 25, 30 and results are in Figure 7(b). We see bounded errors become larger with larger batch size. This is consistent with existing observations (Geiping et al., 2020) on empirical evaluations.

**Error bounds on closed-form DRAs:** Our theoretical results can be also applied in closed-form attacks. Here, we choose the Robbing attack (Fowl et al., 2022) for evaluation and its details are in Appendix D.1.2. The results for single image and batch images recovery on the three datasets and two FL algorithms are shown in Figure 5 and 11. We can see Robbing obtains both small empirical errors and bounded errors (which are even smaller than GGL). This is because its equation solving is suitable to linear layers, and hence relatively accurate on the federated $\ell_2$-LogReg and federated 2-LinConvNet models.

**Practical use of our error bound:** Our error bound has several benefits in practice. First, it can provide guidance or insights to attackers, e.g., the attack performance can be estimated with our error bound prior to the attacks. Then, attackers can make better decisions on the attacks, e.g., try to improve the attack beyond the worst-case bound (using different settings), choose to not perform the attack if the estimated error is too high. Second, it can understand the least efforts to guide the defense design, e.g., when we pursue a minimum obfuscation of local data/model (maintain high utility) and minimum defense based on worst-case attack performance could be acceptable in practice.

**Gap between theoretical bounds & empirical results:** The bounds discussed in the paper represent the worst-case errors, inherent to a theoretical analysis, while the empirical results shown is the average outcome from our experiments. In Section 4.2.1, we point out that the correlation between the two is high, indicating a strong consistency in the relationship between the theoretical bounds and the experimental results while the gap persists. The bounds we have shown reliably rank attack effectiveness relative to one another, which is their primary purpose. For example, the tighter bound for InvGrad over DLG aligns with its superior empirical performance, validating the framework's utility for comparative analysis.

**Convex-loss assumption and practical relevance:** Extending our analysis to non-convex losses (e.g., deep neural networks) is an important direction. Our choice to focus on convex losses was motivated by the need for tractable theoretical guarantees, which provide foundational insights into attack effectiveness. While

real-world FL often uses non-convex objectives, our framework establishes a baseline for understanding how reconstruction errors propagate in a simpler setting. We acknowledge this limitation and emphasize it as a critical area for future work.

## 6 Related Work

Existing DRAs to FL are roughly classified as optimization based and close-form based.

**Optimization-based DRAs to FL:** A series of works (Hitaj et al., 2017; Zhu et al., 2019; Wang et al., 2019; Zhao et al., 2020; Wei et al., 2020; Yin et al., 2021; Jeon et al., 2021; Dang et al., 2021; Balunovic et al., 2022; Sun et al., 2021; Fowl et al., 2022; Wen et al., 2022; Li et al., 2022; Wang et al., 2023) formulate DRAs as the *gradient matching* problem, i.e., an optimization problem that minimizes the difference between gradient from the raw data and that from the reconstructed counterpart. Some works found the gradient itself includes insufficient information to well recover the data (Jeon et al., 2021; Zhu & Blaschko, 2021). For example, Zhu & Blaschko (2021) show there exist pairs of data (called twin data) that visualize different, but have the same gradient. To mitigate this issue, a few works propose to incorporate prior knowledge (e.g., total variation (TV) regularization (Geiping et al., 2020; Yin et al., 2021), batch normalization (BN) statistics (Yin et al., 2021)) into the training data, or introduce an auxiliary dataset to simulate the training data distribution (Hitaj et al., 2017; Wang et al., 2019; Jeon et al., 2021) (e.g., via generative adversarial networks (GANs) Goodfellow et al. (2014)). Though empirically effective, these methods are less practical or data inefficient, e.g., TV is limited to natural images, BN statistics are often unavailable, and training an extra model requires a large amount of data.

**Closed-form DRAs to FL:** A few recent works  (Geiping et al., 2020; Zhu & Blaschko, 2021; Fowl et al., 2022) derive closed-form solutions to reconstruct data, but they require the neural network used in the FL algorithm be fully connected (Geiping et al., 2020), linear/ReLU Fowl et al. (2022), or convolutional (Zhu & Blaschko, 2021).

We will design a framework to theoretically understand the DRA to FL in a general setting, and provide a way to compare the effectiveness of existing DRAs.

## 7 Conclusion

Federated learning (FL) is vulnerable to Data Reconstruction Attacks where an adversary leaks all information about the input data. Existing attacks mainly enhance the empirical attack performance, but lack a theoretical understanding. We study DRAs to FL from a theoretical perspective. Our theoretical results provide a unified way to compare existing attacks theoretically. We also validate our theoretical results via evaluations on multiple datasets and baseline attacks. Future works include: 1) designing better Lipschitz estimation algorithms to obtain tighter error bounds; 2) generalizing our theoretical results to non-convex losses; and 3) designing *theoretically* better DRAs (i.e., with smaller Lipschitz) as well as effective defenses against the attacks (i.e., ensuring larger Lipschitz of their reconstruction function), inspired by our framework.

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

---

**Algorithm 4** Optimization-based data reconstruction attacks (e.g., DLG, iDLG, InvGrad, and GGL)

---

**Input:** Model parameters $\mathbf{w}_t$; true gradient $g(\mathbf{x}, y)$; $\eta, \lambda$; public generator $G(\cdot)$, transformation operator $\mathcal{T}$.
**Output:** Reconstructed data $\hat{\mathbf{x}}$.

 1: **if DLG then**
 2:     $\mathbf{x}'_0 \sim \mathcal{N}(0, 1)$, $y'_0 \sim \mathcal{N}(0, 1)$
 3: **else**
 4:     **if GGL then**
 5:         $\mathbf{z}'_0 \sim \mathcal{N}(0, \mathbf{I}_u)$;
 6:     **else**
 7:         $\mathbf{x}'_0 \sim \mathcal{N}(0, 1)$ // Initialize dummy input(s)
 8:     **end if**
 9:     Estimate $y$ as $\hat{y}$ via methods in Zhao et al. (2020) for a single input or Yin et al. (2021) for a batch inputs
10: **end if**
11: **for** $i = 0; i < I; i + +$ **do**
12:     **if DLG then**
13:         $g(\mathbf{x}'_i, y'_i) \leftarrow \nabla_{\mathbf{w}} \mathcal{L}(\mathbf{w}_t; (\mathbf{x}'_i, y'_i))$
14:         $\mathrm{GML}_i \leftarrow \|g(\mathbf{x}, y) - g(\mathbf{x}'_i, y'_i)\|_2^2$
15:         $\mathbf{x}'_{i+1} \leftarrow \mathbf{x}'_i - \eta \nabla_{\mathbf{x}'_i} \mathrm{GML}_i$
           $y'_{i+1} \leftarrow y'_i - \eta \nabla_{y'_i} \mathrm{GML}_i$
16:     **else if iDLG then**
17:         $g(\mathbf{x}'_i, \hat{y}) \leftarrow \nabla_{\mathbf{w}} \mathcal{L}(\mathbf{w}_t; (\mathbf{x}'_i, \hat{y}))$
18:         $\mathrm{GML}_i \leftarrow \|g(\mathbf{x}, y) - g(\mathbf{x}'_i, \hat{y})\|_2^2$
19:         $\mathbf{x}'_{i+1} \leftarrow \mathbf{x}'_i - \eta \nabla_{\mathbf{x}'} \mathrm{GML}_i$
20:     **else if InvGrad then**
21:         $g(\mathbf{x}'_i, \hat{y}) \leftarrow \nabla_{\mathbf{w}} \mathcal{L}(\mathbf{w}_t; (\mathbf{x}'_i, \hat{y}))$
22:         $\mathrm{GML}_i \leftarrow 1 - \frac{\langle g(\mathbf{x}, y), g(\mathbf{x}'_i, \hat{y})\rangle}{\|g(\mathbf{x}, y)\|_2 \cdot \|g(\mathbf{x}'_i, \hat{y}\|_2}$
23:         $\mathbf{x}'_{i+1} \leftarrow \mathbf{x}'_i - \eta \nabla_{\mathbf{x}'_i} \left(\mathrm{GML}_i + \lambda \mathrm{Reg}_{\mathrm{TV}}(\mathbf{x}'_i)\right)$
24:     **else if GGL then**
25:         $\mathbf{x}'_i = G(\mathbf{z}'_i)$
26:         $g(\mathbf{x}'_i, \hat{y}) \leftarrow \nabla_{\mathbf{w}} \mathcal{L}(\mathbf{w}_t; (\mathbf{x}'_i, \hat{y}))$
27:         $\mathrm{GML}_i \leftarrow \|g(\mathbf{x}, y) - \mathcal{T}(g(\mathbf{x}'_i, \hat{y}))\|_2^2$
28:         $\mathrm{Reg}(G, \mathbf{z}'_i) = (\|\mathbf{z}_i\|_2^2 - k)^2$
29:         $\mathbf{z}'_{i+1} \leftarrow \mathbf{z}'_i - \eta \nabla_{\mathbf{z}'_i} \left(\mathrm{GML}_i + \lambda \mathrm{Reg}(G, \mathbf{z}'_i)\right)$
30:     **end if**
31:     $\mathbf{x}'_{i+1} = \max(\mathbf{x}'_{i+1}, 0)$
32: **end for**
    **Return** $\mathbf{x}'_I$ or $G(\mathbf{z}'_I)$

---

# A   Assumptions

To ensure FedAvg guarantees to converge to the global optimal, existing works have the following assumptions on the local devices' loss functions $\{\mathcal{L}_k\}$.

**Assumption A.1.** $\{\mathcal{L}_k\}'s$ are $L$-smooth: $\mathcal{L}_k(\mathbf{v}) \leq \mathcal{L}_k(\mathbf{w}) + (\mathbf{v} - \mathbf{w})^T \nabla \mathcal{L}_k(\mathbf{w}) + \frac{L}{2}\|\mathbf{v} - \mathbf{w}\|_2^2, \forall \mathbf{v}, \mathbf{w}$.

**Assumption A.2.** $\{\mathcal{L}_k\}'s$ are $\mu$-strongly convex: $\mathcal{L}_k(\mathbf{v}) \geq \mathcal{L}_k(\mathbf{w}) + (\mathbf{v} - \mathbf{w})^T \nabla \mathcal{L}_k(\mathbf{w}) + \frac{\mu}{2}\|\mathbf{v} - \mathbf{w}\|_2^2, \forall \mathbf{v}, \mathbf{w}$.

**Assumption A.3.** Let $\xi_t^k$ be sampled from the $k$-th device's data uniformly at random. The variance of stochastic gradients in each device is bounded: $\mathbb{E}\left\|\nabla \mathcal{L}_k(\mathbf{w}_t^k, \xi_t^k) - \nabla \mathcal{L}_k(\mathbf{w}_t^k)\right\|^2 \leq \sigma_k^2, \forall k$.

**Assumption A.4.** The expected squared norm of stochastic gradients is uniformly bounded, i.e., $\mathbb{E}\left\|\nabla \mathcal{L}_k(\mathbf{w}_t^k, \xi_t^k)\right\|^2 \leq G^2, \forall k, t$.

Note that Assumptions A.1 and A.2 are generic. Typical examples include regularized linear regression, logistic regression, softmax classifier, and recent convex 2-layer ReLU networks (Pilanci & Ergen, 2020). For instance, the original FL (McMahan et al., 2017) uses a 2-layer ReLU networks. Assumptions A.3 and A.4

are used by the existing works (Stich, 2018; Stich et al., 2018; Yu et al., 2019; Li et al., 2020) to derive the convergence condition of FedAvg to reach the global optimal. Note that the loss of deep neural networks is often non-convex, i.e., do not satisfy Assumption A.2. We acknowledge it is important future work to generalize our theoretical results to more challenging non-convex losses.

## B   Proof of Theorem 3.2 for Full Device Participation

Our proof is mainly inspired by the proofs in Stich (2018); Yu et al. (2019); Li et al. (2020).

**Notations:** Let $N$ be the total number of user devices and $K(\leq N)$ be the maximal number of devices that participate in every communication round. Let $T$ be the total number of every device's SGDs, and $E$ be the number of each device's local updates between two communication rounds. Thus $T/E$ is the number of communications, assuming $E$ is dividable by $T$.

Let $\mathbf{w}_t^k$ be the model parameter maintained in the $k$-th device at the $t$-th step. Let $\mathcal{I}_E$ be the set of global aggregation steps, i.e., $\mathcal{I}_E = \{nE \mid n = 1, 2, \cdots\}$. If $t + 1 \in \mathcal{I}_E$, i.e., the devices communicate with the server and the server performs the `FedAvg` aggregation on device models. Then the update of `FedAvg` with partial devices active can be described as

$$\mathbf{v}_{t+1}^k = \mathbf{w}_t^k - \eta_t \nabla \mathcal{L}_k(\mathbf{w}_t^k, \xi_t^k), \tag{6}$$

$$\mathbf{w}_{t+1}^k = \begin{cases} \mathbf{v}_{t+1}^k & \text{if } t + 1 \notin \mathcal{I}_E, \\ \sum_{k=1}^N p_u \mathbf{v}_{t+1}^k & \text{if } t + 1 \in \mathcal{I}_E. \end{cases} \tag{7}$$

Motivated by (Stich, 2018; Li et al., 2020), we define two virtual sequences $\mathbf{v}_t = \sum_{k=1}^N p_k \mathbf{v}_t^k$ and $\mathbf{w}_t = \sum_{k=1}^N p_k \mathbf{w}_t^k$. $\mathbf{v}_{t+1}$ results from an single step of SGD from $\mathbf{w}_t$. When $t + 1 \notin \mathcal{I}_E$, both are inaccessible. When $t + 1 \in \mathcal{I}_E$, we can only fetch $\mathbf{w}_{t+1}$. For convenience, we define $\overline{\mathbf{g}}_t = \sum_{k=1}^N p_k \nabla \mathcal{L}_k(\mathbf{w}_t^k)$ and $\mathbf{g}_t = \sum_{k=1}^N p_k \nabla \mathcal{L}_k(\mathbf{w}_t^k, \xi_t^k)$. Hence, $\mathbf{v}_{t+1} = \mathbf{w}_t - \eta_t \mathbf{g}_t$ and $\mathbb{E}\mathbf{g}_t = \overline{\mathbf{g}}_t$.

Before proving Theorem 3.2, we need below key lemmas that are from Stich (2018); Li et al. (2020).

**Lemma B.1** (Results of one step SGD). *Assume Assumptions A.1 and A.2 hold. If $\eta_t \leq \frac{1}{4L}$, we have*

$$\mathbb{E}\left\|\mathbf{v}_{t+1} - \boldsymbol{w}^\star\right\|^2 \leq (1 - \eta_t\mu)\mathbb{E}\left\|\boldsymbol{w}_t - \boldsymbol{w}^\star\right\|^2 + \eta_t^2 \mathbb{E}\left\|\mathbf{g}_t - \overline{\mathbf{g}}_t\right\|^2 + 6L\eta_t^2\Gamma + 2\mathbb{E}\sum_{k=1}^N p_k \left\|\boldsymbol{w}_t - \boldsymbol{w}_k^t\right\|^2$$

*where $\Gamma = \mathcal{L}^* - \sum_{k=1}^N p_k \mathcal{L}_k^\star \geq 0$.*

*Proof sketch:* Lemma B.1 is mainly from Lemma 1 in Li et al. (2020). The proof idea is to bound three terms, i.e., the inner product $\langle \mathbf{w}_t - \mathbf{w}^*, \nabla \mathcal{L}(\mathbf{w}_t)\rangle$, the square norm $||\nabla \mathcal{L}(\mathbf{w}_t)||^2$, and the inner product $\langle \nabla \mathcal{L}_k(\mathbf{w}_t), \mathbf{w}_t^k - \mathbf{w}_t\rangle, \forall k$. Then, the left-hand term in Lemma B.1 can be rewritten in terms of the three terms and be bounded by the right-hand four terms in Lemma B.1. Specifically, 1) It first bounds $\langle \mathbf{w}_t - \mathbf{w}^*, \nabla \mathcal{L}(\mathbf{w}_t)\rangle$ using the strong convexity of the loss function (Assumption A.2); 2) It bounds $||\nabla \mathcal{L}(\mathbf{w}_t)||^2$ using the smoothness of the loss function (Assumption A.1); and 3) It bounds $\langle \nabla \mathcal{L}_k(\mathbf{w}_t), \mathbf{w}_t^k - \mathbf{w}_t\rangle, \forall k$ using the convexity of the loss function (Assumption A.2).

**Lemma B.2** (Bounding the variance). *Assume Assumption A.3 holds. Then $\mathbb{E}\left\|\mathbf{g}_t - \overline{\mathbf{g}}_t\right\|^2 \leq \sum_{k=1}^N p_u^2 \sigma_u^2$.*

**Lemma B.3** (Bounding the divergence of $\{\mathbf{w}_t^k\}$). *Assume Assumption A.4 holds, and $\eta_t$ is non-increasing and $\eta_t \leq 2\eta_{t+E}$ for all $t \geq 0$. It follows that $\mathbb{E}\left[\sum_{k=1}^N p_k \left\|\boldsymbol{w}_t - \boldsymbol{w}_k^t\right\|^2\right] \leq 4\eta_t^2(E-1)^2G^2$.*

Now, we complete the proof of Theorem 3.2.

*Proof.* First, we observe that no matter whether $t+1 \in \mathcal{I}_E$ or $t+1 \notin \mathcal{I}_E$ in Equation (7), we have $\mathbf{w}_{t+1} = \mathbf{v}_{t+1}$. Denote $\Delta_t = \mathbb{E}\left\|\mathbf{w}_t - \mathbf{w}^\star\right\|^2$. From Lemmas B.1 to B.3, we have

$$\Delta_{t+1} = \mathbb{E}\left\|\mathbf{w}_{t+1} - \mathbf{w}^\star\right\|^2 = \mathbb{E}\left\|\mathbf{v}_{t+1} - \mathbf{w}^\star\right\|^2 \leq (1 - \eta_t\mu)\Delta_t + \eta_t^2 B \tag{8}$$

where $B = \sum_{k=1}^{N} p_u^2 \sigma_u^2 + 6L\Gamma + 8(E-1)^2 G^2$.

For a diminishing stepsize, $\eta_t = \frac{\beta}{t+\gamma}$ for some $\beta > \frac{1}{\mu}$ and $\gamma > 0$ such that $\eta_1 \leq \min\{\frac{1}{\mu}, \frac{1}{4L}\} = \frac{1}{4L}$ and $\eta_t \leq 2\eta_{t+E}$. We will prove $\Delta_t \leq \frac{v}{\gamma+t}$ where $v = \max\left\{\frac{\beta^2 B}{\beta\mu-1}, (\gamma+1)\Delta_1\right\}$.

We prove it by induction. Firstly, the definition of $v$ ensures that it holds for $t = 1$. Assume the conclusion holds for some $t$, it follows that

$$
\begin{aligned}
\Delta_{t+1} &\leq (1 - \eta_t \mu)\Delta_t + \eta_t^2 B \\
&\leq \left(1 - \frac{\beta\mu}{t+\gamma}\right)\frac{v}{t+\gamma} + \frac{\beta^2 B}{(t+\gamma)^2} \\
&= \frac{t+\gamma-1}{(t+\gamma)^2}v + \left[\frac{\beta^2 B}{(t+\gamma)^2} - \frac{\beta\mu-1}{(t+\gamma)^2}v\right] \\
&\leq \frac{v}{t+\gamma+1}.
\end{aligned}
$$

By the $\bar{L}$-Lipschitz continuous property of $\text{Rec}(\cdot)$,

$$\|\text{Rec}(\mathbf{w}_t) - \text{Rec}(\mathbf{w}^*)\| \leq \bar{L} \cdot \|\mathbf{w}_t - \mathbf{w}^\star\|.$$

Then we have

$$\mathbb{E}\|\text{Rec}(\mathbf{w}_t) - \text{Rec}(\mathbf{w}^*)\|^2 \leq \bar{L}^2 \cdot \mathbb{E}\|\mathbf{w}_t - \mathbf{w}^\star\|^2 \leq \bar{L}^2 \Delta_t \leq \bar{L}^2 \frac{v}{\gamma+t}.$$

Specifically, if we choose $\beta = \frac{2}{\mu}, \gamma = \max\{8\frac{L}{\mu}, E\} - 1$, then $\eta_t = \frac{2}{\mu}\frac{1}{\gamma+t}$. We also verify that the choice of $\eta_t$ satisfies $\eta_t \leq 2\eta_{t+E}$ for $t \geq 1$. Then, we have

$$v = \max\left\{\frac{\beta^2 B}{\beta\mu-1}, (\gamma+1)\Delta_1\right\} \leq \frac{\beta^2 B}{\beta\mu-1} + (\gamma+1)\Delta_1 \leq \frac{4B}{\mu^2} + (\gamma+1)\Delta_1.$$

Hence,

$$\mathbb{E}\|\text{Rec}(\mathbf{w}_t) - \text{Rec}(\mathbf{w}^*)\|^2 \leq \bar{L}^2 \frac{v}{\gamma+t} \leq \frac{\bar{L}^2}{\gamma+t}\left(\frac{4B}{\mu^2} + (\gamma+1)\Delta_1\right).$$

$\square$

## C    Proofs of Theorem 3.3 for Partial Device Participation

Recall that $\mathbf{w}_t^k$ is $k$-th device's model at the $t$-th step, $\mathcal{I}_E = \{nE \mid n = 1, 2, \cdots\}$ is the set of global aggregation steps; $\bar{\mathbf{g}}_t = \sum_{k=1}^{N} p_k \nabla \mathcal{L}_k(\mathbf{w}_t^k)$ and $\mathbf{g}_t = \sum_{k=1}^{N} p_k \mathcal{L}_k(\mathbf{w}_t^k, \xi_t^k)$, and $\mathbf{v}_{t+1} = \mathbf{w}_t - \eta_t \mathbf{g}_t$ and $\mathbb{E}\mathbf{g}_t = \bar{\mathbf{g}}_t$. We denote by $\mathcal{H}_t$ the multiset selected which allows any element to appear more than once. Note that $\mathcal{H}_t$ is only well defined for $t \in \mathcal{I}_E$. For convenience, we denote by $\mathcal{S}_t = \mathcal{H}_{N(t,E)}$ the most recent set of chosen devices where $N(t, E) = \max\{n|n \leq t, n \in \mathcal{I}_E\}$.

In partial device participation, FedAvg first samples a random multiset $\mathcal{S}_t$ of devices and then only performs updates on them. Directly analyzing on the $\mathcal{S}_t$ is complicated. Motivated by Li et al. (2020), we can use a thought trick to circumvent this difficulty. Specifically, we assume that `FedAvg` always activates *all devices* at the beginning of each round and uses the models maintained in only a few sampled devices to produce the next-round model. It is clear that this updating scheme is equivalent to that in the partial device participation. Then the update of `FedAvg` with partial devices activated can be described as:

$$\mathbf{v}_{t+1}^k = \mathbf{w}_t^k - \eta_t \nabla \mathcal{L}_k(\mathbf{w}_t^k, \xi_t^k), \tag{9}$$

$$\mathbf{w}_{t+1}^k = \begin{cases} \mathbf{v}_{t+1}^k & \text{if } t+1 \notin \mathcal{I}_E, \\ \text{samples } \mathcal{S}_{t+1} \text{ and average } \{\mathbf{v}_{t+1}^k\}_{k \in \mathcal{S}_{t+1}} & \text{if } t+1 \in \mathcal{I}_E. \end{cases} \tag{10}$$

Note that in this case, there are two sources of randomness: stochastic gradient and random sampling of devices. The analysis for Theorem 3.2 in Appendix B only involves the former. To distinguish with it, we use an extra notation $\mathbb{E}_{\mathcal{S}_t}(\cdot)$ to consider the latter randomness.

First, based on Li et al. (2020), we have the following two lemmas on unbiasedness and bounded variance. Lemma C.1 shows the scheme is unbiased, while Lemma C.2 shows the expected difference between $\mathbf{v}_{t+1}$ and $\mathbf{w}_{t+1}$ is bounded.

**Lemma C.1** (Unbiased sampling scheme). *If $t + 1 \in \mathcal{I}_E$, we have $\mathbb{E}_{\mathcal{S}_t}(\boldsymbol{w}_{t+1}) = \mathbf{v}_{t+1}$.*

**Lemma C.2** (Bounding the variance of $\mathbf{w}_t$). *For $t + 1 \in \mathcal{I}$, assume that $\eta_t$ is non-increasing and $\eta_t \leq 2\eta_{t+E}$ for all $t \geq 0$. Then the expected difference between $\mathbf{v}_{t+1}$ and $\boldsymbol{w}_{t+1}$ is bounded by*

$$\mathbb{E}_{\mathcal{S}_t} \|\mathbf{v}_{t+1} - \boldsymbol{w}_{t+1}\|^2 \leq \frac{4}{K}\eta_t^2 E^2 G^2.$$

Now, we complete the proof of Theorem 3.3.

*Proof.* Note that

$$\begin{aligned}
\|\mathbf{w}_{t+1} - \mathbf{w}^*\|^2 &= \|\mathbf{w}_{t+1} - \mathbf{v}_{t+1} + \mathbf{v}_{t+1} - \mathbf{w}^*\|^2 \\
&= \underbrace{\|\mathbf{w}_{t+1} - \mathbf{v}_{t+1}\|^2}_{A_1} + \underbrace{\|\mathbf{v}_{t+1} - \mathbf{w}^*\|^2}_{A_2} + \underbrace{2\langle \mathbf{w}_{t+1} - \mathbf{v}_{t+1}, \mathbf{v}_{t+1} - \mathbf{w}^*\rangle}_{A_3}.
\end{aligned}$$

When expectation is taken over $\mathcal{S}_{t+1}$, the last term ($A_3$) vanishes due to the unbiasedness of $\mathbf{w}_{t+1}$.

If $t + 1 \notin \mathcal{I}_E$, $A_1$ vanishes since $\mathbf{w}_{t+1} = \mathbf{v}_{t+1}$. We use Lemma C.2 to bound $A_2$. Then it follows that

$$\mathbb{E} \|\mathbf{w}_{t+1} - \mathbf{w}^*\|^2 \leq (1 - \eta_t\mu)\mathbb{E} \|\mathbf{w}_t - \mathbf{w}^\star\|^2 + \eta_t^2 B.$$

If $t + 1 \in \mathcal{I}_E$, we additionally use Lemma C.2 to bound $A_1$. Then

$$\begin{aligned}
\mathbb{E} \|\mathbf{w}_{t+1} - \mathbf{w}^*\|^2 &= \mathbb{E} \|\mathbf{w}_{t+1} - \mathbf{v}_{t+1}\|^2 + \mathbb{E} \|\mathbf{v}_{t+1} - \mathbf{w}^*\|^2 \\
&\leq (1 - \eta_t\mu)\mathbb{E} \|\mathbf{w}_t - \mathbf{w}^\star\|^2 + \eta_t^2 B + \frac{4}{K}\eta_t^2 E^2 G^2 \\
&= (1 - \eta_t\mu)\mathbb{E} \|\mathbf{w}_t - \mathbf{w}^\star\|^2 + \eta_t^2 (B + C),
\end{aligned} \tag{11}$$

where $C = \frac{4}{K}E^2 G^2$ is the upper bound of $\frac{1}{\eta_t^2}\mathbb{E}_{\mathcal{S}_t} \|\mathbf{v}_{t+1} - \mathbf{w}_{t+1}\|^2$.

We observe that the only difference between equation 11 and equation 8 is the additional $C$. Thus we can use the same argument there to prove the theorems here. Specifically, for a diminishing stepsize, $\eta_t = \frac{\beta}{t+\gamma}$ for some $\beta > \frac{1}{\mu}$ and $\gamma > 0$ such that $\eta_1 \leq \min\{\frac{1}{\mu}, \frac{1}{4L}\} = \frac{1}{4L}$ and $\eta_t \leq 2\eta_{t+E}$, we can prove $\mathbb{E} \|\mathbf{w}_{t+1} - \mathbf{w}^*\|^2 \leq \frac{v}{\gamma+t}$ where $v = \max\left\{\frac{\beta^2(B+C)}{\beta\mu-1}, (\gamma+1)\|\mathbf{w}_1 - \mathbf{w}^*\|^2\right\}$.

Then by the $\bar{L}$-Lipschitz continuous property of $\text{Rec}(\cdot)$,

$$\mathbb{E}\|\text{Rec}(\mathbf{w}_t) - \text{Rec}(\mathbf{w}^*)\|^2 \leq \bar{L}^2 \cdot \mathbb{E} \|\mathbf{w}_t - \mathbf{w}^\star\|^2 \leq \bar{L}^2\Delta_t \leq \bar{L}^2\frac{v}{\gamma+t}.$$

Specifically, if we choose $\beta = \frac{2}{\mu}, \gamma = \max\{8\frac{L}{\mu}, E\} - 1$,

$$\mathbb{E}\|\text{Rec}(\mathbf{w}_t) - \text{Rec}(\mathbf{w}^*)\|^2 \leq \bar{L}^2\frac{v}{\gamma+t} \leq \frac{\bar{L}^2}{\gamma+t}\left(\frac{4(B+C)}{\mu^2} + (\gamma+1)\|\mathbf{w}_1 - \mathbf{w}^*\|^2\right).$$

$\square$

# D Experiments

## D.1 More Experimental Setup

### D.1.1 Details about the FL algorithms and unrolled feed-forward network

We first show how to compute calculate $L, \mu, \sigma_u$, and $G$ in Assumptions 1-4 on federated $\ell_2$-regularized logistic regression ($\ell_2$-LogReg) and federated 2-layer linear convolutional network (2-LinConvNet); Then we show how to compute the Lipschitz $L_{\mathcal{R}}$ on each data reconstruction attack.

**Federated $\ell_2$-LogReg:** Each device $k$'s local objective is $\mathcal{L}_k(\mathbf{w}) = \frac{1}{\bar{n}} \sum_{j=1}^{\bar{n}} \log(1 + \exp(-y_j \langle \mathbf{w}, \mathbf{x}_j^k \rangle)) + \gamma \|\mathbf{w}\|^2$. In our results, we simply set $\gamma = 0.1$ for brevity.

- **Compute** $L$: first, all $\mathcal{L}_k$'s are $\frac{1}{4}(\frac{1}{\bar{n}} \sum_j \|x_j^k\|^2)$-smooth (Papailiopoulos, 2018); then $L = \max_{k \in [N]} \frac{1}{4}(\frac{1}{\bar{n}} \sum_j \|\mathbf{x}_j^k\|^2) + 2\gamma$;

- **Compute** $\mu$: all $\mathcal{L}_k$'s are $\gamma$-strongly convex for the $\gamma$ regularized $\ell_2$ logistic regression (Papailiopoulos, 2018) and $\mu = \gamma$.

- **Compute** $\sigma^k$ **and** $G$: we first traverse all training data $\xi_t^k$ in the $k$-th device in any $t$-th round and then use them to calculate the maximum square norm differences $\left\| \nabla \mathcal{L}_k(w_t^k, \xi_t^k) - \nabla \mathcal{L}_k(w_t^k) \right\|^2$. Similarly, $G$ can be calculated as the maximum value of the expected square norm $\left\| \nabla \mathcal{L}_k(w_t^k, \xi_t^k) \right\|^2$ among all devices $\{k\}$ and rounds $\{t\}$.

**Federated 2-LinConvNet (Pilanci & Ergen, 2020).** Let a two-layer network $f : \mathbb{R}^d \to \mathbb{R}$ with $m$ neurons be: $f(\mathbf{x}) = \sum_{j=1}^{m} \phi(\mathbf{x}^T \mathbf{u}_j) \alpha_j$, where $\mathbf{u}_j \in \mathbb{R}^d$ and $\alpha_j \in \mathbb{R}$ are the weights for hidden and output layers, and $\phi(\cdot)$ is an activation function. Two-layer convolutional networks with $U$ filters can be described by patch matrices (e.g., images) $\mathbf{X}_u, u = 1, \cdots, U$. For flattened activations, we have $f(\mathbf{X}_1, \cdots \mathbf{X}_u) = \sum_{u=1}^{U} \sum_{j=1}^{m} \phi(\mathbf{X}_u \mathbf{u}_j) \alpha_j$.

We consider the 2-layer linear convolutional networks and its non-convex loss is defined as:

$$\min_{\{\alpha_j, \mathbf{u}_j\}_{j=1}^m} \mathcal{L}(\{\alpha_j, \mathbf{u}_j\}) = \frac{1}{2} \| \sum_{u=1}^{U} \sum_{j=1}^{m} \mathbf{X}_u \mathbf{u}_j \alpha_{ju} - \mathbf{y} \|_2^2. \tag{12}$$

Pilanci & Ergen (2020) show that the above non-convex problem can be transferred to the below convex optimization problem via its duality. and the two problems have the identical optimal values:

$$\min_{\{\mathbf{w}_u \in \mathbb{R}^d\}_{u=1}^U} \mathcal{L}(\{\mathbf{w}_u\}) = \frac{1}{2} \| \sum_{u=1}^{U} \mathbf{X}_u \mathbf{w}_u - \mathbf{y} \|_2^2. \tag{13}$$

We run federated learning with convex 2-layer linear convolutional network, where each device trains the local loss $\mathcal{L}_k(\{\mathbf{w}_u\}_{u=1}^U)$ and it can converge to the optimal model $\mathbf{w}^* = \{\mathbf{w}_u^*\}$.

- **Compute** $L$: Let $\underline{\mathbf{w}} = \{\mathbf{w}_u\}_{u=1}^U$. For each client $k$, we require its local loss $\mathcal{L}_k$ should satisfy $\|\nabla \mathcal{L}_k(\underline{\mathbf{w}}) - \nabla \mathcal{L}_k(\underline{\mathbf{v}})\|_2 \le L_k \|\underline{\mathbf{w}} - \underline{\mathbf{v}}\|_2$ for any $\underline{\mathbf{w}}, \underline{\mathbf{v}}$; With Equation 13, we have

$$\left\| \sum_{u=1}^{U} (\mathbf{X}_u^k)^T \mathbf{X}_u^k (\underline{\mathbf{w}} - \underline{\mathbf{v}}) \right\|_2 \le L_k \|\underline{\mathbf{w}} - \underline{\mathbf{v}}\|_2$$

; Then we have the smoothness constant $L_k$ to be the maximum eigenvalue of $\sum_{u=1}^{U} (\mathbf{X}_u^k)^T \mathbf{X}_u^k$, which is $\|\sum_{u=1}^{U} (\mathbf{X}_u^k)^T \mathbf{X}_u^k\|_2$; Hence, $L = \max_k \|\sum_{u=1}^{U} (\mathbf{X}_u^k)^T \mathbf{X}_u^k\|_2$.

- **Compute** $\mu$: Similar as computing $L$, $\mu$ is the min eigenvalue of $\sum_{u=1}^{U} (\mathbf{X}_u^k)^T \mathbf{X}_u^k$ for all $k$, that is, $\mu = \min_k \|\sum_{u=1}^{U} (\mathbf{X}_u^k)^T \mathbf{X}_u^k\|_2$.

Table 1: Best empirical errors (log scale) of the baseline attacks on the three datasets in the default setting.

| Data/Algorithm | Fed. $\ell_2$-LogReg: single | | | | Fed. 2-LinConvNet: single | | | | Fed. $\ell_2$-LogReg: batch | | | |
|---|---|---|---|---|---|---|---|---|---|---|---|---|
| | DLG | iDGL | InvGrad | GGL | DLG | iDGL | InvGrad | GGL | DLG | iDGL | InvGrad | GGL |
| **MNIST** | 3.48 | 3.38 | 3.09 | 3.14 | 3.69 | 3.50 | 3.42 | 3.29 | 6.38 | 6.28 | 6.08 | 4.94 |
| **FMNIST** | 3.54 | 3.42 | 3.35 | 3.25 | 3.84 | 3.58 | 3.52 | 3.13 | 6.44 | 6.35 | 6.29 | 4.98 |
| **CIFAR10** | 4.32 | 4.13 | 4.13 | 3.92 | 4.52 | 3.37 | 4.33 | 3.70 | 6.46 | 6.45 | 6.63 | 5.31 |

- **Compute $\sigma^k$ and $G$**: Similar computation as in $\ell_2$-regularized logistic regression.

**Unrolled feed-forward network and its training and performance.** In our experiments, we set the number of layers to be 20 in the unrolled feed-forward network for the three datasets. We use 1,000 data samples and their intermediate reconstructions to train the network. To reduce overfitting, we use the greedy layer-wise training strategy. For instance, the average MSE (between the input and output of the unrolled network) of DLG, iDLG, InvGrad, and GGL on MNIST is 1.22, 1.01, 0.76, and 0.04, respectively—indicating that the training performance is promising. After training the unrolled network, we use the AutoLip algorithm to calculate the Lipschitz $L_{\mathcal{R}}$.

### D.1.2   Details about Data Reconstruction Attacks

**GGL (Li et al., 2022)**: GGL considers the scenario where clients realize the server will infer their private data and they hence perturb their local models before sharing them with the server as a defense. To handle noisy models, GGL solves an optimization problem similar to InvGrad, but uses a pretrained generator as a regularization. The generator is trained on the entire MNIST dataset and can calibrate the reconstructed noisy image to be within the image manifold. Specifically, given a well-trained generator $G(\cdot)$ on public datasets and assume the label $y$ is inferred by iGLD, GGL targets the following optimization problem:

$$\mathbf{z}^* = \arg \min_{\mathbf{z} \in \mathbb{R}^k} \mathrm{GML}(g(\mathbf{x}, y), \mathcal{T}(g(G(\mathbf{z}), y))) + \lambda \mathrm{Reg}(G; \mathbf{z}), \tag{14}$$

where $\mathbf{z}$ is the latent space of the generative model, $\mathcal{T}$ is a lossy transformation (e.g., compression or sparsification) acting as a defense, and $\mathrm{Reg}(G; \mathbf{z})$ is a regularization term that penalizes the latent $\mathbf{z}$ if it deviates from the prior distribution. Once the optimal $\mathbf{z}^*$ is obtained, the image can be reconstructed as $G(\mathbf{z}^*)$ and should well align the natural image.

In the experiments, we use a public pretrained GAN generator for MNIST, Fashion-MNIST, and CIFAR. We adopt gradient clipping as the defense strategy $\mathcal{T}$ performed by the clients. Specifically, $\mathcal{T}(g, S) = g/\max(1, \|g\|_2/S)$. Note that since $G(\cdot)$ is trained on the whole image dataset, it produces stable reconstruction during the optimization.

**Robbing (Fowl et al., 2022)**: Robbing approximately reconstructs the data via solving an equation without any iterative optimization. Assume a batch of data $\mathbf{x}_1, \mathbf{x}_2, \cdots \mathbf{x}_n$ with unique labels $\mathbf{y}_1, \mathbf{y}_2, \cdots \mathbf{y}_n$ in the form of one-hot encoding. Let $\oslash$ be element-wise division. Then, Robbing observes that each row $i$ in $\frac{\partial \mathcal{L}_t}{\partial y_t}$, i.e., $\frac{\partial \mathcal{L}_t}{\partial y_t^i}$, actually recovers

$$\mathbf{x}_t = \frac{\partial \mathcal{L}_t}{\partial y_t^i} \mathbf{x}_t \oslash \frac{\partial \mathcal{L}_t}{\partial y_t^i}.$$

In other others, Robbing directly maps the model to the reconstructed data. Hence, in our experiment, the unrolled feed-forward neural network reduces to 1-layer ReLU network. We then estimate Lipschitz upper bound on this network.

### D.2   More Experimental Results

Table 1 show the *best* one-snapshot empirical results of the four attacks on the two FL algorithms in the default setting.

Figures 8-10 show the batch images recovery results by the four considered DRAs on federated $\ell_2$-LogReg.

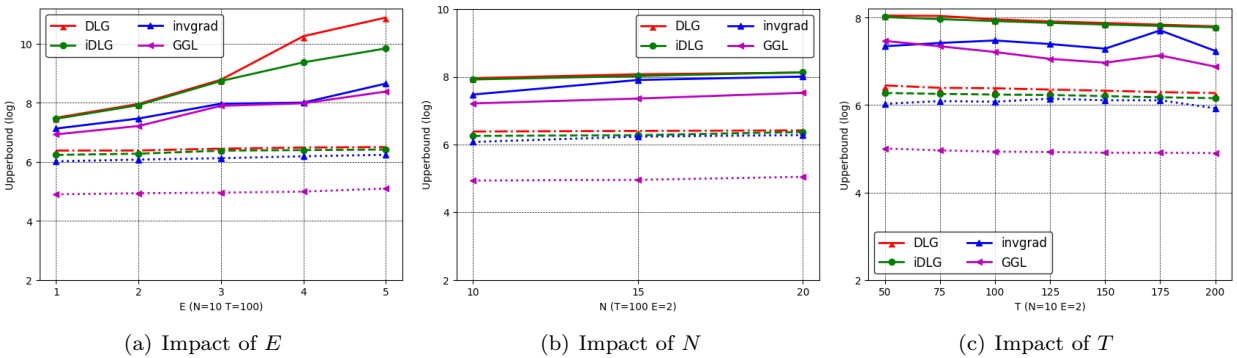

Figure 8: Results of federated $\ell_2$-LogReg on MNIST—batch images recovery.

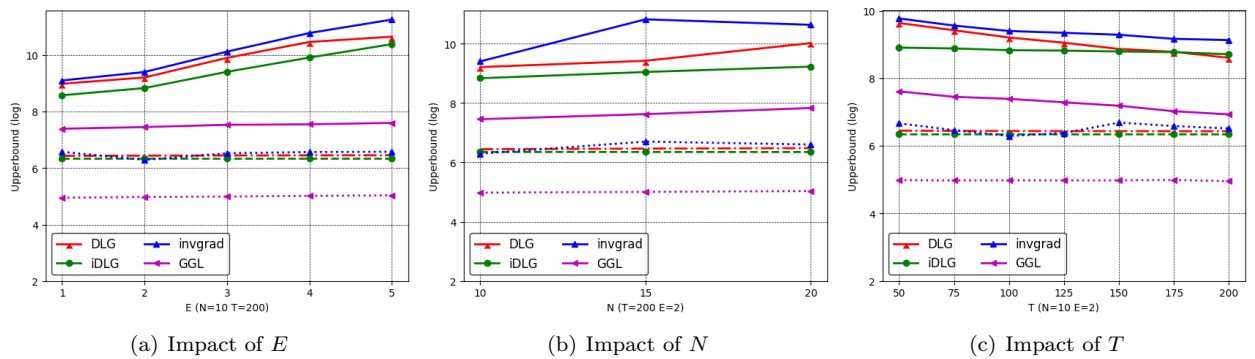

Figure 9: Results of federated $\ell_2$-LogReg on FMNIST—batch images recovery.

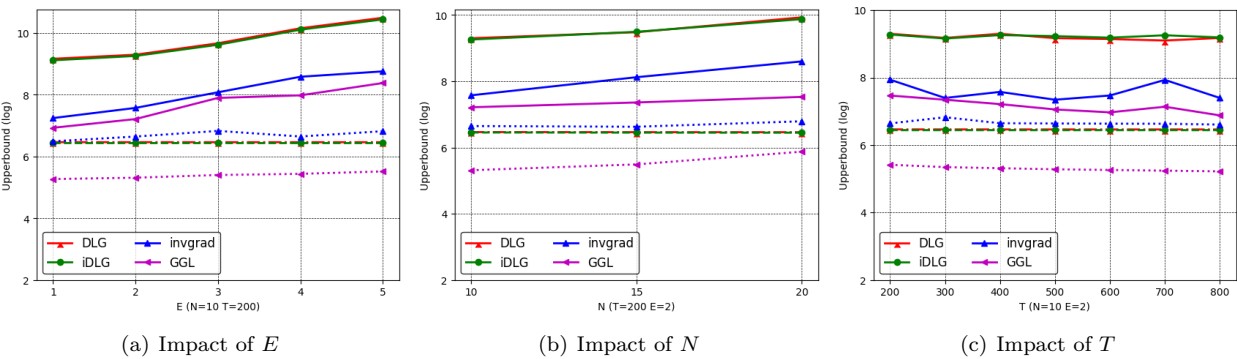

Figure 10: Results of federated $\ell_2$-LogReg on CIFAR10—batch images recovery.

Figure 11 shows the batch image recovery results by Robbing on federated 2-LinConvNet.

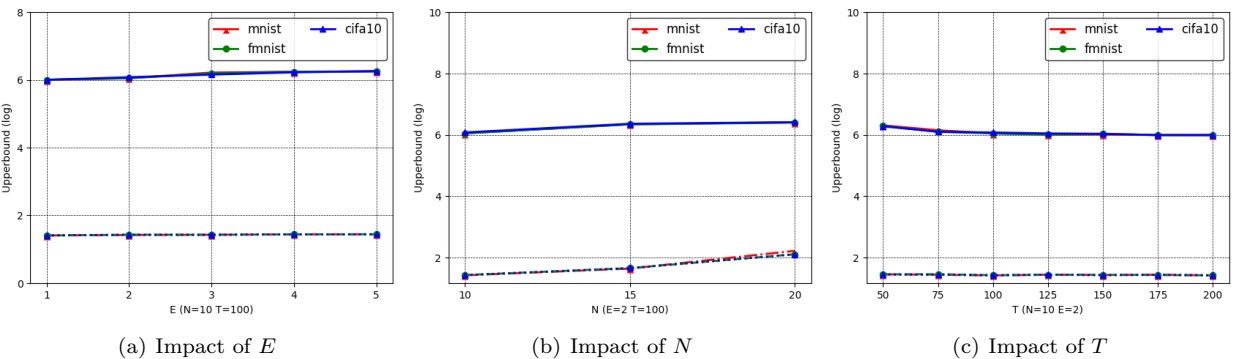

(a) Impact of $E$  (b) Impact of $N$  (c) Impact of $T$

Figure 11: Results of federated $\ell_2$-LogReg on Robbing—batch images recovery.

