# OpenReview forum: "Theoretically Understanding Data Reconstruction Leakage in Federated Learning"
_TMLR — Accepted by TMLR_

### Review · Reviewer_ZkRn · 2025-11-17

**Summary Of Contributions:**

The paper introduces a theoretical framework to bound the data reconstruction error in the context of Federated Learning.
The result is relevant and useful for theoretically compare the the effectiveness of existing Data Reconstruction Attacks (DRA) by comparing their bounded errors.
The framework is tested on multiple attacks and on three different datasets (Mnist, Fashion-Mnist and Cifar-10) under the assumption of having a convex loss (which is probably the main weakness of the paper).

**Audience:**

Yes

**Audience Explanation:**

I believe the contribution is sound and that it could be beneficial for researchers working in the field.

**Broader Impact Concerns:**

I do not have any ethical concerns about this work

**Claims And Evidence:**

Yes

**Claims Explanation:**

The claims made in the submission are convincing, but they are based on a strong assumption of the convex loss. The authors recognize this limitation on page 10 and also leave the extension to non-convex losses as future work.
The second thing that could improve the contribution is a study with more clients involved in the FL model training to simulate a more "real-life" scenario.

**Requested Changes:**

Main problems:

1) The abstract is not clear and does not clearly explain what is the contribution of the paper. In the end, the names of iDLG and DLG are introduced, but at that point, a reader that do not already know them does not clearly know what they are, and most importantly, if they are a contribution of the paper.

2) The authors provided experiments with a relatively low number of clients (between 5 and 15). Would the proposed approach work in a cross-device case in which the number of clients is higher (let's say 100 clients) and only a few of the clients are selected in each FL round? Could you please elaborate a bit on this or conduct some experiments showing how a higher number of clients can impact the results?

3) The main limitation is the assumption of the convex loss. This is also one of the future works. Could the authors elaborate a bit about how difficult it would be to extend the framework to a more standard non-convex loss scenario?

4) The plots can be improved and are not easy to read. Some suggestions for all the plots of the paper:
- In all the plots, the font of x, y labels, ticks, and legend should be increased because it is hard to read them
- In Figures 3, 4, 5, 6, and 7, besides the labels and ticks that are hard to read, I would suggest using the same y limit in the plots regarding the same dataset. For instance, Figures 6a, b, and c should have all 10 as a maximum y limit.
- Since the legend is the same for all the plots, it could be placed once below the plots, increasing the font size. Moreover, I should also include the meaning of the dashed lines in the legend.

Minor comments:

- There are some typos and other minor problems with grammar and text, I've found some of them that I listed here
- In the Introduction, the server is referred to as "center server" while in another point of the paper as "central server". Usually, in FL papers, it is called "central server". I would suggest using this when referring to the server to be more consistent in the different parts of the paper.
- On page 2, when FedAvg is introduced, it is mentioned that clients "only share the gradients". This is how FedSGD works, with FedAvg clients sharing their locally trained models so that they can run multiple local updates.
- Page 3: "aims to infers" -> "aims to infer"
- Page 4: "Give random data" -> "Given random data"
- In the abstract, there is a repetition of the word "attack": an attack’s error bound reflects its inherent attack effectiveness.
- In the introduction "keep and train their data locally" -> train their model not data
- There is a missing word here "Third, the error bounds do not consistent correlations"
- I don't know if it is a formatting error but in the abstract there are a couple of missing spaces between works like "attacks.However"

---

### Review · Reviewer_asth · 2025-11-21

**Summary Of Contributions:**

This paper proposes a theoretical framework to bound data reconstruction error in FL. The authors introduce a metric based on the Lipschitz constant of the reconstruction function to quantify attack effectiveness, applying this theory to validate that iDLG inherently outperforms DLG.

**Additional Comments:**

## Minor comments

- Abstract/Introduction: FL is referred to as an "emerging" field - it has been around for nearly a decade
- Section 1: "Hence, Attack A shows....effective than B" : as the adversary you don't care about "truly more effective". If some attack can exploit certain snapshots very efficiently, that's a win for the adversary. What one should care about is worst-case across the entire training run.
- The paper has a lot of math and theorems, but it is not apparent how they are useful. For instance, Theorems 3.1 through 3.3 involve a lot of (overly complicated) equations, but it is not clear to me what their impact is supposed to be in terms of the story of the paper, or how they are helpful in conclusions that follow?
- Section 3.2: "... followed by a clip nonlinear activation function.." - a) please be specific about the activation function b) why is this particular activation function added?
- Section 5: 'First term vs second term in the error bound' : the exact values are not very useful here for a discussion, since the average reader would not have good intuition about what a values for "the two terms in DLG" mean.
- Page 2: "Our theoretical results show that when an attacker’s reconstruction function has a smaller Lipschitz constant, then this attack intrinsically performs better" - A function that always predict 0 or some small value $\epsilon$ technically has a very small Lipschitz constant, but is of no use to any adversary. The authors here should clarify that the smaller constant is more useful for robustness of reconstruction to noise.

**Audience:**

Yes

**Audience Explanation:**

- Data reconstruction in the context of FL may be of interest to certain researchers focusing on connecting theory and practice for privacy analyses
- Working towards ways to predict worst-case reconstruction for attacks can be beneficial, especially when trying to establish the superiority of one method over another by reporting more than just empirical reconstruction attack results.

**Claims And Evidence:**

No

**Claims Explanation:**

- While the authors acknowledge the convex-loss assumption as a limitation, as a theoretical paper it should be more than just an acknowledgement. This is a very strong assumption that is likely to never hold for models trained with SGD in FL (or any other setting). Even if the theoretical analyses are limited to convex losses, the authors should experiment with models trained in a way that would make their loss curves non-convex, and see how well the theoretical results here hold.
- The reported error bounds (Figures 3, 4) are multiple orders of magnitude worse than actual empirical bounds (note that the y-axis is on the log scale), suggesting that the bounds are too loose and may not be of practical use. The authors should either work on making the bounds tighter, or discuss how the bounds are useful despite them being significantly off from empirical observations.
- "Third, the error bounds do not consistent correlations with the best empirical errors" (Page 9) is a direct contradiction to the central claim of the paper (that theoretical bounds can be useful in understanding/prediction attack success). Correlation with "average" performance is not useful under an adversarial setting- all involved parties in a security context care about worst-case risk, not average-care risk
- Page 5: The way FL devices are sampled here per round is inconsistent with standard FL-  devices per round are sampled without replacement, but the text right above Theorem 3m3 mensions sampling "independently and with replacement"

**Requested Changes:**

- Figure 1: Figure takes up too much blank space. What message is the figure supposed to convey? The way the MSE is visualized here is very odd (separate lines for mean and std) - please replace with just one line, along with 1 (or 2) standard error worth of shaded region. On that note, given the high MSEs I don't know if the difference in values for the two attacks is statistically significant
- Equation 2 focuses on the gradient of an individual record, but as an adversary you only get to observe batch gradients (since each client runs SGD locally on data in batches). Even in that case, the adversary has no way of knowing 'which' batch might have the target data in it. It is unclear to me how these constraints are incorporated in the attack, and it would be great if the authors could comment on this and clarify.
- The derivation in Equation 3 is unclear to me: the expectation here is supposed to be over the randomness of R, but the text accompanying this equation comments on R being a constant.
- Page 7: "...we use a public pretrained GAN generator for ..."  - do you mean a) some generator model X, which is kept the same for all three datasets and publicly available, used or b) a separate generator model X_i for each dataset (with the generator trained on that particular dataset)? Please clarify, since the latter implies that the adversary somehow has indirect access to the very dataset over which it seeks to perform reconstruction
- I do not agree with the suggested practical use of error bounds (page 10) - the reconstruction attack is passive i.e. as an adversary, there is no harm in doing anyway, and always trying to improve performance within reasonable measure is also reasonable for an adversary.

---

### Review · Reviewer_KxVA · 2025-11-29

**Summary Of Contributions:**

This paper proposes a theoretical framework for analyzing gradient-based data reconstruction attacks (DRAs) in federated learning. The authors model such attacks as a reconstruction function $R(w_t)$ of the global model and derive reconstruction error upper bounds under convex loss assumptions for FedAvg with both full and partial client participation. Empirically, the authors show, across several datasets and FL configurations, that the resulting bounds correlate well with average empirical reconstruction error and capture the effects of key FL hyperparameters, thereby suggesting that the framework is useful as a ranking and comparison tool for DRAs.

**Additional Comments:**

I don't have additional comments.

**Audience:**

No

**Audience Explanation:**

* Typos

The paper has numerous typographical errors and missing symbols. This makes the technical claims difficult to verify. Please see details in the previous section.


* The assumption of convexity has some limitations.

The paper assumes that the objective loss functions are convex. Under this setting, the empirical results show that the average reconstruction error is strongly correlated with the worst-case error bound derived by the authors. However, it is unclear whether a similar correlation holds in the non-convex setting, which is common in federated learning. This limitation restricts the scope of the paper. Section 5 (paragraph 5) includes some discussion of the worst-case upper bound, which is intended to inform the difficulty of data reconstruction attacks from the perspective of an adversary. However, it remains unclear how the insights gained from the convex setting can be extended to more practical, non-convex scenarios.

Given these two concerns, I believe that the current version is not yet ready for publication in TMLR.

**Broader Impact Concerns:**

No ethical concerns.

**Claims And Evidence:**

No

**Claims Explanation:**

I noticed several potential typographical errors and instances of missing symbols in the paper. I just listed a few examples:

* The meaning of $||_1$ is unclear in equations (4) and (5)
* the $t k$ in the last two lines of page 14
* the $t k$ in the line below equation (8)
* the $t=t$, $t=t k$ in fourth-to-last lines on page 15
* the $t-*$ in the final line of page 15
* the $t-tk$ in Lemma C.3
* the $-*$ in equation (9)
* Many more typos on pages 16, 17, and 18.

Due to these typographical errors, the correctness of the technical claims is difficult to verify.

**Requested Changes:**

(1) I would recommend that the authors carefully revise these typos, as doing so would significantly enhance the clarity of the technical content and make the paper’s contributions more accessible to the readership.

(2) Could the authors elaborate on the practical implications of this work? In particular, how do the conclusions derived under the convexity assumption inform the design of attacks or defenses in federated learning systems with non-convex objectives? It would be helpful to clarify how the insights from the convex setting generalize (or fail to generalize) to more realistic non-convex settings, as the current discussion in Section 5 (paragraph 5) doesn't provide these details.

---

### Comment · Action_Editor_Thpz · 2025-12-14
**Rebuttal**

Dear Authors,

This is a kind reminder that the rebuttal period has started for a while and we haven’t received your response. Are you planning to submit a rebuttal? The reviewers will be asked to submit their final recommendations soon. Thanks!

Best,
AE

---

### Author Response · Authors · 2025-12-14
**Reply to Reviewers (Rebuttlal)**

Typos, Minor Comments and Formatting: We have resolved the LaTeX compilation issues, and the correct mathematical expressions will be displayed in the revised version. We will modify the abstract to improve clarity, correct the portion that mistakenly referred to FedSGD as FedAvg, fix typos in the report, and address the remaining minor comments in the revision.

Reviewer KxVA:

Response to Convexity Assumption: We acknowledge that our analysis assumes convex loss functions. However, we emphasize that several practical FL algorithms indeed utilize convex objectives, including ones used in our experiments. While real-world FL often involves non-convex objectives, our framework establishes a necessary baseline for understanding reconstruction errors in these tractable settings. Extending these theoretical guarantees to non-convex losses is an important direction for future work.

Reviewer asth:

Refer to part I for convexity assumption response.

Response to Statistical Significance and Figure 1: The reviewer’s observation accurately reflects our intended message. The high MSE makes it difficult to conclude statistical significance via empirical analysis alone; this instability is exactly what we aimed to convey in Figure 1. We will add a clarification in the third paragraph of the Introduction explicitly mentioning Figure 1 to emphasize this point.

Response to Correlation (Section 4.2.1): Regarding the comment on Page 9 about the correlation between the best empirical error and the theoretical error bound, this observation was specific to the best single image recovery. Section 4.2.1 marks our observations for single image recovery, which is highly sensitive to initialization and thus more unstable than batch recovery. This instability necessitates a theoretical bound to provide a reliable baseline.

Response to Partial Device Participation: The reviewer correctly notes that we adopt an FL scenario with partial device participation and replacement. This aligns with the work of Li et al. (2020), where convergence results for partial participation are derived under this assumption. We adopt replacement here to maintain tractability in the theoretical analysis of convergence.

Response to Gap between Empirical and Theoretical Results: It is correct to observe a significant gap between the empirical results and the theoretical bound. It is worth noting that a hidden "Ideal Reconstruction Error" lies between these two numbers; our theoretical bound approaches this from above (worst-case), while empirical results approach it from below. Crucially, despite the numerical gap, we observe a strong correlation between our theoretical bound and the empirical outcome of these attacks on batch recoveries. The utility of our approach lies in providing a theoretically backed method to compare reconstruction strength (ranking models), rather than relying solely on empirical results which are sensitive to initialization.

Response to Batch Gradients and Target Data: The reviewer is correct that the adversary only observes batch gradients. In our scenario, consistent with standard FL Data Reconstruction Attack (DRA) settings, there is no specific "target" data point. All data contributing to the shared gradients are considered private. Therefore, successfully reconstructing any data sample from a client's batch constitutes a successful attack. We will add this clarification to Section 2.2.

Response to Equation 3 (Expectation): We apologize for the confusion caused by the phrasing. To clarify, the expectation in Equation 3 is indeed taken over the randomness inherent in the reconstruction function (e.g., random initialization of dummy data/labels), as noted in Section 3.1. However, regarding the ideal model R(w*) (the optimal global model learned by the FL algorithm), the associated reconstruction term is considered constant as it does not depend on the initialization.

Response to GGL Generator Setup: On Page 7 (Experiment Setup 4.1), our GGL setup strictly adheres to the protocol outlined in Section 3.3 of the original GGL paper (Li et al., 2022). As mentioned in Section 4.1 (Implementation), the FL task is performed on the evaluation set, which is disjoint from the GAN training set. We will add further clarification in our revision to address potential confusion regarding the dataset split.

Response to Practical Use: We will clarify our suggestion on Page 10. We intend to convey that an attacker can leverage the bound to decide how many resources to invest in tuning the attack or when to modify the attack approach.

Reviewer ZkRn:

Response to Impact of N: Regarding the impact of the number of clients N, our theoretical results (Theorems 3.2 and 3.3) show that a larger N incurs a larger error bound. This is consistent with the intuition that a larger N makes FL training more unstable and harder to converge, thereby increasing the reconstruction error bound.

Refer to part I for convexity assumption response.

---

### Decision · Action_Editor_Thpz · 2026-01-11

**Recommendation:** Accept as is

**Audience:**

Yes

**Audience Explanation:**

The findings are of interest to the TMLR audience, particularly researchers focused on Privacy-Preserving Machine Learning and Federated Learning security.

**Claims And Evidence:**

Yes

**Claims Explanation:**

This paper presents a theoretical framework for analyzing Data Reconstruction Attacks in Federated Learning. The authors derive a bound for reconstruction error and propose using the Lipschitz constant of the reconstruction function as a metric to quantify the inherent effectiveness of an attack. The study theoretically and empirically compares existing attacks, specifically validating that iDLG inherently outperforms DLG. The analysis focuses on convex loss functions to establish a baseline for understanding reconstruction leakage.

The submission's claims are supported by clear evidence. The authors have derived theoretical bounds for reconstruction errors in FL settings and validated their theoretical ranking of attacks through empirical experiments on MNIST, Fashion-MNIST, and CIFAR-10. The reviewers agreed that the theoretical framework correlates well with the empirical results regarding attack effectiveness.

All three reviewers share a final recommendation of Leaning Accept. Reviewers highlighted the theoretical framework for comparing attacks and the successful validation of the theory against empirical error rates. The primary limitation cited by all reviewers is the convexity assumption for the loss function.